# Sequential Probability Assignment with Contexts: Minimax Regret, Contextual Shtarkov Sums, and Contextual Normalized Maximum Likelihood

**Ziyi Liu**
University of Toronto & Vector Institute
`kevind.liu@mail.utoronto.ca`

**Idan Attias**
Ben-Gurion University & Vector Institute
`idanatti@post.bgu.ac.il`

**Daniel M. Roy**
University of Toronto & Vector Institute
`daniel.roy@utoronto.ca`

## Abstract

We study the fundamental problem of sequential probability assignment, also known as online learning with logarithmic loss, with respect to an arbitrary, possibly nonparametric hypothesis class. Our goal is to obtain a complexity measure for the hypothesis class that characterizes the minimax regret and to determine a general, minimax optimal algorithm. Notably, the sequential $\ell_\infty$ entropy, extensively studied in the literature (Rakhlin and Sridharan, 2015, Bilodeau et al., 2020, Wu et al., 2023), was shown to not characterize minimax regret in general. Inspired by the seminal work of Shtarkov (1987) and Rakhlin, Sridharan, and Tewari (2010), we introduce a novel complexity measure, the *contextual Shtarkov sum*, corresponding to the Shtarkov sum after projection onto a multiary context tree, and show that the worst case log contextual Shtarkov sum equals the minimax regret. Using the contextual Shtarkov sum, we derive the minimax optimal strategy, dubbed *contextual Normalized Maximum Likelihood* (cNML). Our results hold for sequential experts, beyond binary labels, which are settings rarely considered in prior work. To illustrate the utility of this characterization, we provide a short proof of a new regret upper bound in terms of sequential $\ell_\infty$ entropy, unifying and sharpening state-of-the-art bounds by Bilodeau et al. (2020) and Wu et al. (2023).

## 1 Introduction

Sequential probability assignment is a fundamental problem with connections to information theory [Ris84; MF98; XB00], machine learning [CL06; Vov95; RST15; FKLMS18; Sha20], and portfolio optimization [Kel56; Cov74; Cov91; CO96; Fed91]. In the original non-contextual setup, the learner aims to assign probabilities to a series of labels, which are revealed sequentially. The goal is to offer probabilistic forecasts over the label set such that the probability assigned to any observed sequence is comparable to that assigned by the best model in any fixed class of models.

The celebrated work of Shtarkov [Sht87] characterized minimax regret for context-free sequential probability assignment in terms of what is now known as the *Shtarkov sum*, and subsequently described the minimax algorithm, *Normalized Maximum Likelihood* (NML). NML represents the ideal probabilistic forecast in the sense of minimax regret, providing a benchmark for universal coding and prediction strategies. While often not used directly due to its computational complexity, NML has guided the design of practical algorithms and informed the development of efficient approxima-

38th Conference on Neural Information Processing Systems (NeurIPS 2024).

tion methods. The principles underlying NML have inspired advances in both information theory and online learning, establishing fundamental limits and serving as a critical benchmark.

In this work, we study the problem of sequential probability assignment with contexts, which has been analyzed in recent works (e.g. [RS15; BFR20; WHGS23]) under the framework of online supervised learning formalized by Rakhlin, Sridharan, and Tewari [RST10]. In this setup, the problem is modeled as a $T$-round game between a *learner* and the *nature*: On each round $t = 1, \ldots, T$, the learner observes a context $x_t$ from nature and predicts a distribution $\hat{p}_t$ over some finite label space $\mathcal{Y}$. Then nature reveals a label $y_t \in \mathcal{Y}$ and the learner incurs a logarithmic loss $\ell(\hat{p}_t, y_t) = -\log(\hat{p}_t(y_t))$, where $\hat{p}_t(y_t)$ is the probability assigned to label $y_t$ by $\hat{p}_t$. The performance of the learner is measured by the *regret* with respect to a class $\mathcal{F}$ of *experts*, defined as the difference between the total loss of the learner and that of the best expert in $\mathcal{F}$. The value of primary interest is the *minimax regret*, that is, the worst-case regret by the best learner over arbitrarily adaptive data sequences. The minimax regret serves as a benchmark for all algorithms and as a target for studies of adaptivity. Our goal is to address several fundamental questions:

*Can we find a natural complexity measure of $\mathcal{F}$ that characterizes the minimax regret, enabling us to analyze the minimax regret in new ways? And can we identify, in view of this complexity measure, a general, minimax optimal algorithm?*

Notably, the sequential covering number of $\mathcal{F}$, a well studied measure of complexity, has been shown not to characterize the minimax regret on its own [RS15; BFR20; WHGS23]. This fact distinguishes sequential probability assignment and log loss: while sequential covering numbers enable a tight analysis in online learning problems with convex Lipschitz losses, like absolute loss [RST15] and square loss [RS14a], they do not yield minimax rates for log loss on some classes. Tackling such classes evidently requires new techniques.

**Main contributions.**

1. We introduce a new complexity measure, which we call the *contextual Shtarkov sum*, that serves as a natural generalization of the Shtarkov sum from the context-free setting. We show that the minimax regret is characterized by the worst-case contextual Shtarkov sum.

2. We derive the minimax optimal algorithm, dubbed *contextual Normalized Maximum Likelihood* (cNML), using a data-dependent variant of the contextual Shtarkov sum, thereby generalizing NML from the context-free setting.

3. We apply contextual Shtarkov sums to the study of sequential entropy bounds on minimax. Doing so, we provide a short proof of a new regret upper bound in terms of sequential entropy that unifies and even *improves* on state-of-the-arts bounds by [BFR20] and [WHGS23]. Our results extend beyond the binary label setting studied by recent work to arbitrary finite label sets.

**Related work.** Sequential probability assignment has been studied extensively. Various aspects of this problem have been investigated, including sequences with and without side information (contexts), parametric and nonparametric hypothesis classes, and stochastic or adversarial data-generating mechanisms. This problem has a long history in the machine learning community, see [CL06, Ch. 9] and the references therein. In the information theory community, this problem is also known as universal prediction [MF98], where the regret is also referred to as redundancy with respect to a set of codes. This has been studied in both stochastic and adversarial settings [Fre96; Ris86; Ris96; Sht87; XB97; DS04; MF98; OS04; Sha06; Szp98], where the focus was primarily on the parametric classes of experts. Closely related topics include universal source coding [Kol65; Sol64; Fit66; Dav73], data compression with arithmetic coding [Ris76; RL81; ZL77; ZL78; FMG92], and the minimum description length (MDL) principle [Ris78; Ris84; Ris87; BRY98; Grü05; HY01]. Lastly, this topic is intimately tied with sequential gambling and portfolio optimization, as pointed out by [Kel56; Cov74; Cov91; CO96; Fed91].

A classical result in context-free sequential probability assignment is that the minimax regret is equal to the log contextual Shtarkov sum [Sht87], and the minimax algorithm is the well-known Normalized Maximum Likelihood. When the set of contexts is known in advance to the forecaster, namely, a fixed design setting, the minimax regret is equivalent to the log Shtarkov sum of the function class when projected onto the known set of contexts [JSS21; WHGS23].

A long line of work has focused on controlling minimax regret for rich classes in terms of covering numbers. [CL99; OH99] upper bounded the regret in terms of the (non-sequential) uniform covering number of the class. However, this complexity measure proved to be insufficient for obtaining optimal rates. [RS15] improved regret upper bounds by proposing a sequential covering measure. Thereafter, by utilizing the self-concordance property and the curvature of the log loss, [BFR20] further improved the upper bound in terms of the sequential covering number for nonparametric Lipschitz classes, through a non-constructive proof. [WHGS23] proposed a Bayesian algorithmic approach in order to upper bound the regret using a global notion of sequential covering. Notably, both the global and local sequential covering numbers do not fully characterize the regret, and the algorithm in [WHGS23] is not minimax optimal. By relaxing the worst-case analysis, [BHS23] studied this problem within the smoothed analysis framework, where nature is not fully adversarial but constrained.

Online learning with respect to arbitrary hypothesis classes and the zero-one loss, in the realizable case, is known to be characterized by the Littlestone dimension [Lit88]. The agnostic case was addressed by [BPS09; ABDMNY21]. Understanding sequential complexities in online learning with Lipschitz losses was extensively studied by [RST10; RS14a; RS14b; RST15]. Note that the logarithmic loss is neither Lipschitz nor bounded. Recently, [AHKKV23] characterized online regression in the realizable case, for any approximate pseudo-metric, such as the $\ell_p$ loss.

## 2   Preliminaries

**Notation.**   For a positive integer $K$, let $[K] := \{1, 2, \ldots, K\}$. For a finite set $\mathcal{K}$ with $|\mathcal{K}| = K$, we use $\Delta(\mathcal{K})$ to denote the set of all distributions on $\mathcal{K}$. We may identify $\mathcal{K}$ with $[K]$ (under arbitrary enumeration of elements in $\mathcal{K}$) and treat elements of $\Delta(\mathcal{K})$ as vectors in $\mathbb{R}^K$. For a vector $p \in \mathbb{R}^K$ and $i \in [K]$, let $p(i)$ be the $i$-th coordinate of $p$. Let $\Delta^+(\mathcal{K}) = \{p \in \Delta(\mathcal{K}) : p(k) > 0, \forall k \in \mathcal{K}\}$. For a general finite sequence $(a_i)_{i=1}^N$, we will use $a_{n:m}$ to denote the sub-sequence $(a_n, \ldots, a_m)$ for any $n \le m$ and the empty sequence for $n > m$. For any set $\mathcal{A}$, let $\mathcal{A}^* = \cup_{k \ge 0} \mathcal{A}^k$ be the set of all finite length sequences over $\mathcal{A}$.

**Sequential probability assignment and minimax regret.**   Let $\mathcal{X}$ be the context space and $\mathcal{Y}$ be the finite label space. In each round $t \in [T]$ during the game of sequential probability assignment, the learner receives a context $x_t \in \mathcal{X}$ from nature and assigns a probability distribution $\hat{p}_t \in \Delta(\mathcal{Y})$ to the possible labels. Then nature reveals the true label $y_t \in \mathcal{Y}$ and the learner incurs a loss $\ell(\hat{p}_t, y_t) = -\log(\hat{p}_t(y_t))$. Throughout, the learner is required to predict nearly as well as the best expert from an expert class, which is modeled as an arbitrary hypothesis class $\mathcal{F} \subseteq \{(\mathcal{X} \times \mathcal{Y})^* \times \mathcal{X} \to \Delta(\mathcal{Y})\}$. More formally, the goal of the learner is make their *regret* with respect to $\mathcal{F}$,

$$\mathcal{R}_T(\mathcal{F}; \hat{p}_{1:T}, x_{1:T}, y_{1:T}) = \sum_{t=1}^T \ell(\hat{p}_t, y_t) - \inf_{f \in \mathcal{F}} \sum_{t=1}^T \ell(f(x_{1:t}, y_{1:t-1}), y_t),$$

as small as possible for all sequences **x** and **y** generated by nature, possibly in an adversarial manner. Here $f(x_{1:t}, y_{1:t-1}) \in \Delta(\mathcal{Y})$ can be understood as the prediction made by expert $f$ at round $t$ using past observations $(x_{1:t-1}, y_{1:t-1})$ as well as the fresh context $x_t$. The main focus is to study the *minimax regret* $\mathcal{R}_T(\mathcal{F})$, which can be written as the following extensive form

$$\mathcal{R}_T(\mathcal{F}) = \sup_{x_1} \inf_{\hat{p}_1} \sup_{y_1} \cdots \sup_{x_T} \inf_{\hat{p}_T} \sup_{y_T} \mathcal{R}_T(\mathcal{F}; \hat{p}_{1:T}, x_{1:T}, y_{1:T}),$$

where $x_t \in \mathcal{X}, \hat{p}_t \in \Delta(\mathcal{Y})$ and $y_t \in \mathcal{Y}, \forall t \in [T]$. In light of this formulation, we can see that the minimax regret concerns both the learner and the nature to be *adaptive*, meaning that their actions can rely on the revealed history so far.

**Remark 2.1 (Sequential vs non-sequential experts)** Experts $f$ as mappings from $(\mathcal{X} \times \mathcal{Y})^* \times \mathcal{X}$ to $\Delta(\mathcal{Y})$ are sometimes called *fully sequential* experts [WHGS23] due to their ability to predict based on the past history. However, the literature (e.g. [RS15; BFR20; WHGS23]) often considers the more limited notion of *non-sequential* experts, modeled as $\mathcal{F} \subseteq \{\mathcal{X} \to \Delta(\mathcal{Y})\}$, reflecting the fact that prediction made by each expert $f$ is simply $f(x_t)$ in each round $t$. In contrast, our results are more general as our novel techniques can be applied to the more flexible sequential experts.

**Multiary trees.** The complexity of online learning problems stems from the sequential and adaptive nature of the adversary, which we can capture with *multiary trees*. Formally, for a general space $\mathcal{A}$ and a finite set $\mathcal{K}$, an $\mathcal{A}$-valued $\mathcal{K}$-ary tree $\mathbf{v}$ of depth $d$ is a sequence of mappings $\mathbf{v}_t : \mathcal{K}^{t-1} \to \mathcal{A}$ for $t \in [d]$. A *path* in a depth-$d$ tree is a sequence $\boldsymbol{\varepsilon} = (\varepsilon_1, \ldots, \varepsilon_d) \in \mathcal{K}^d$. We use the notation $\mathbf{v}_t(\boldsymbol{\varepsilon})$ to denote $\mathbf{v}_t(\varepsilon_1, \ldots, \varepsilon_{t-1})$ for $t \in [d]$ and the boldface notation $\mathbf{v}(\boldsymbol{\varepsilon})$ to denote $(\mathbf{v}_1(\boldsymbol{\varepsilon}), \ldots, \mathbf{v}_d(\boldsymbol{\varepsilon})) \in \mathcal{A}^d$. Throughout we will only consider $\mathcal{Y}$-ary trees valued in either $\mathcal{X}$ or $\Delta(\mathcal{Y})$, where the paths are denoted by the boldface $\mathbf{y}$. We refer to $\mathcal{X}$-valued trees as *context trees* and $\Delta(\mathcal{Y})$-valued trees as *probabilistic trees*.

**Time-varying context sets.** So far we consider the context set $\mathcal{X}$ to be constant over time. But all of our results can be extended easily to allow for time-varying context spaces. Details of this generalization can be found in Appendix C.

## 2.1 Prior work: the Shtarkov sum in context-free and fixed designs

Before introducing our complexity measure that characterizes $\mathcal{R}_T(\mathcal{F})$, we review some prior settings where the minimax regret can be characterized by the well-studied *Shtarkov sum*. First we introduce the notion of likelihood of a hypothesis $f$ with respect to a context and label sequence, which plays a key role in defining complexity measures and optimal algorithms.

**Definition 2.2 (Likelihood)** For $f : (\mathcal{X} \times \mathcal{Y})^* \times \mathcal{X} \to \Delta(\mathcal{Y})$ and length$-d$ sequences $x_{1:d} \in \mathcal{X}^d, y_{1:d} \in \mathcal{Y}^d$, the likelihood $P_f(y_{1:d}|x_{1:d})$ is defined as

$$P_f(y_{1:d}|x_{1:d}) = \prod_{t=1}^d f(x_{1:t}, y_{1:t-1})(y_t),$$

where we use the compact notation $f(x_{1:t}, y_{1:t-1})$ for $f(x_1, y_1, \ldots, x_{t-1}, y_{t-1}, x_t)$.

In the classical context-free setting where $\mathcal{X}$ can be thought of as a singleton, any sequential expert $f$ degenerates to a joint distribution over label sequences. Indeed, given any label sequence $y_{1:t-1}$, $f(y_{1:t-1}) \in \Delta(\mathcal{Y})$ can be interpreted as the conditional distribution $f$ assigns to the next label $y_t$. We use $P_f(y_{1:d}) = \prod_{t=1}^d f(y_{1:t-1})(y_t)$ to denote this distribution. Similarly, the learner's strategy is also specified by a joint distribution that is decomposed to a sequence of conditional distributions $\hat{p}_t = \hat{p}_t(\cdot|y_{1:t-1}) \in \Delta(\mathcal{Y})$. In this setup the minimax regret $\mathcal{R}_T(\mathcal{F})$ is characterized by the Shtarkov sum [Sht87].

**Proposition 2.3 ([Sht87])** *In the context-free setting, for any hypothesis class $\mathcal{F}$ and horizon $T$, the Shtarkov sum $S_T(\mathcal{F})$ is defined as*

$$S_T(\mathcal{F}) = \sum_{y_{1:T} \in \mathcal{Y}^T} \sup_{f \in \mathcal{F}} P_f(y_{1:T}).$$

*Moreover, the minimax regret is given by $\mathcal{R}_T(\mathcal{F}) = \log S_T(\mathcal{F})$, and the unique minimax optimal strategy is the* normalized maximum likelihood *(NML) distribution given by*

$$p_{nml}(y_{1:T}) = \frac{\sup_{f \in \mathcal{F}} P_f(y_{1:T})}{\sum_{y'_{1:T} \in \mathcal{Y}^T} \sup_{f \in \mathcal{F}} P_f(y'_{1:T})}, \qquad \forall y_{1:T} \in \mathcal{Y}^T.$$

To go beyond this classical context-agnostic setting and incorporate contextual information, prior work (e.g. [JSS21]) also considered an easier problem than the aforementioned sequential probability assignment, by forcing nature to reveal the context sequence $x_{1:T}$ to the learner at the start of the game. This is known as the *fixed design* setting or *transductive online learning* [WHGS23], where the goal is to characterize the so-called *fixed design maximal* minimax regret

$$\mathcal{R}_T^{\mathrm{FD}}(\mathcal{F}) := \sup_{x_{1:T} \in \mathcal{X}^T} \inf_{\hat{p}_1} \sup_{y_1} \cdots \inf_{\hat{p}_T} \sup_{y_T} \mathcal{R}_T(\mathcal{F}; \hat{p}_{1:T}, x_{1:T}, y_{1:T}).$$

It is straightforward to see that after projecting on $x_{1:T}$, the hypothesis class $\mathcal{F}$ again collapses to a set of joint distributions over $\mathcal{Y}^T$ specified by the likelihood function in Definition 2.2. Moreover, this set of distributions can be accessed by the learner from the start, so the fixed design setting can

be essentially reduced to the context-free setting. To be more specific, for any $f \in \mathcal{F}$, it induces an expert in the context-free setting after being projected on $x_{1:T}$, which is denoted by $f|_{x_{1:T}}$ and

$$f|_{x_{1:T}}(y_{1:t-1}) := f(x_{1:t}, y_{1:t-1}) \in \Delta(\mathcal{Y}), \forall t \in [T], y_{1:t-1} \in \mathcal{Y}^{t-1},$$

and let $\mathcal{F}|_{x_{1:T}} := \{f|_{x_{1:T}} : f \in \mathcal{F}\}$. Then given any predetermined $x_{1:T}$, the learner is equivalently competing with $\mathcal{F}|_{x_{1:T}}$ in the context-free setting. With the following natural variant of the Shtarkov sum, we can easily characterize $\mathcal{R}_T^{\mathrm{FD}}(\mathcal{F})$.

**Definition 2.4 (Conditional Shtarkov sum)** Given a context sequence $x_{1:T} \in \mathcal{X}^T$, the Shtarkov sum of $\mathcal{F}$ conditioned on $x_{1:T}$ is

$$S_T(\mathcal{F}|x_{1:T}) := \sum_{y_{1:T} \in \mathcal{Y}^T} \sup_{f \in \mathcal{F}} P_f(y_{1:T}|x_{1:T}).$$

In fact, $S_T(\mathcal{F}|x_{1:T})$ is just the Shtarkov sum of the projected class $\mathcal{F}|_{x_{1:T}}$ in the context-free setting. The following result characterizes the fixed-design setting:

**Proposition 2.5 (Minimax regret, fixed design [JSS21])** *In the fixed design setting, for any hypothesis class $\mathcal{F}$ and horizon $T$, the fixed design maximal minimax regret is*

$$\mathcal{R}_T^{\mathrm{FD}}(\mathcal{F}) = \sup_{x_{1:T} \in \mathcal{X}^T} \log S_T(\mathcal{F}|x_{1:T}),$$

*and, given any context sequence $x_{1:T}$, the minimax optimal response is NML with respect to $\mathcal{F}|_{x_{1:T}}$.*

## 3 Minimax regret via contextual Shtarkov sum

Now we state one of our main results about the characterization of the minimax regret of sequential probability assignment. First we introduce the key concept of *contextual Shtarkov sum*, which is a natural generalization of Shtarkov sum in the context-free setting.

**Definition 3.1 (Contextual Shtarkov sum)** The *contextual Shtarkov sum* $S_T(\mathcal{F}|\mathbf{x})$ of a hypothesis class $\mathcal{F}$ on a given context tree $\mathbf{x}$ of depth $T$ is defined as

$$S_T(\mathcal{F}|\mathbf{x}) := \sum_{\mathbf{y} \in \mathcal{Y}^T} \sup_{f \in \mathcal{F}} P_f(\mathbf{y}|\mathbf{x}(\mathbf{y})).$$

Just like the conditional Shtarkov sum, the contextual Shtarkov sum $S_T(\mathcal{F}|\mathbf{x})$ can be interpreted as the Shtarkov sum of the projected class $\mathcal{F}|_{\mathbf{x}} := \{f|_{\mathbf{x}} : f \in \mathcal{F}\}$ where $f|_{\mathbf{x}}$ is the induced context-free expert specified by

$$f|_{\mathbf{x}}(y_{1:t-1}) := f(\mathbf{x}(y_{1:t-1}), y_{1:t-1}) \in \Delta(\mathcal{Y}), \forall t \in [T], y_{1:t-1} \in \mathcal{Y}^{t-1},$$

where we have slightly abused the notation to use $\mathbf{x}(y_{1:t-1})$ to denote the length-$t$ context sequence obtained by tracing tree $\mathbf{x}$ through the (partial) path $y_{1:t-1}$. Next we show that the minimax regret $\mathcal{R}_T(\mathcal{F})$ is characterized by the worst-case contextual Shtarkov sum:

**Theorem 3.2 (Main result: minimax regret)** *For any hypothesis class $\mathcal{F} \subseteq \{(\mathcal{X} \times \mathcal{Y})^* \times \mathcal{X} \to \Delta(\mathcal{Y})\}$ and horizon $T$,*
$$\mathcal{R}_T(\mathcal{F}) = \sup_{\mathbf{x}} \log S_T(\mathcal{F}|\mathbf{x}),$$

*where the supremum is taken over all $\mathcal{X}$-valued context trees $\mathbf{x}$ of depth $T$.*

Since any context sequence $x_{1:T}$ can be thought as a special context tree $\mathbf{x}$ that is constant in each level $t \in [T]$ (i.e., $\mathbf{x}_t(\mathbf{y}) = x_t, \forall \mathbf{y}$), we can find that the supremum over context trees in Theorem 3.2 strictly subsumes the supremum over context sequences in Proposition 2.5. Thus we can see the separation between $\mathcal{R}_T(\mathcal{F})$ and $\mathcal{R}_T^{\mathrm{FD}}(\mathcal{F})$ is clearly exhibited.

The full proof of Theorem 3.2 as well as an overview are provided in Appendix A.

## 3.1 Applications: an improved regret upper bound in terms of sequential entropy

To illustrate the utility of our characterization in Theorem 3.2, we walk through some examples where we are able to recover and *sharpen* existing regret upper bounds with relatively short proofs via contextual Shtarkov sum. As a start, we provide a short proof in Appendix A.6 of the classical regret bound for a finite hypothesis class.

**Proposition 3.3 (Finite classes)** *For any $\mathcal{F} \subseteq [0,1]^{\mathcal{X}}$ and horizon $T$, $\mathcal{R}_T(\mathcal{F}) \leq \log |\mathcal{F}|$.*

Let us go back to the binary label setting with non-sequential experts, that is, $\mathcal{Y} = \{0,1\}$ and $\mathcal{F} \subseteq [0,1]^{\mathcal{X}}$, and $f(x) \in [0,1]$ is interpreted as the probability assigned to label 1 by this expert $f$. We will show a regret bound that *outperforms* the state-of-the-art ones in [BFR20; WHGS23] with a surprisingly simple proof. To proceed, we need the following notation. Given a context tree $\mathbf{x}$ of depth $T$, let $\mathcal{F} \circ \mathbf{x} = \{f \circ \mathbf{x} : f \in \mathcal{F}\}$, where $f \circ \mathbf{x}$ is the $[0,1]$-valued tree such that

$$(f \circ \mathbf{x})_t(\mathbf{y}) = f(\mathbf{x}_t(\mathbf{y})), \forall \mathbf{y} \in \mathcal{Y}^T.$$

Next we introduce the definitions of sequential $\ell_\infty$ covers and entropy.

**Definition 3.4 (Sequential $\ell_\infty$ cover and entropy)** Given a hypothesis class $\mathcal{F} \subseteq [0,1]^{\mathcal{X}}$ and a context tree $\mathbf{x}$ of depth $T$, we say a collection of $\mathbb{R}$-valued trees $V_{\mathbf{x},\alpha}$ is a sequential cover of $\mathcal{F} \circ \mathbf{x}$ at scale $\alpha > 0$ if for any $f \in \mathcal{F}, \mathbf{y} \in \mathcal{Y}^T$, there exists some $v \in V_{\mathbf{x},\alpha}$ such that

$$|f(\mathbf{x}_t(\mathbf{y})) - v_t(\mathbf{y})| \leq \alpha, \forall t \in [T].$$

Let the sequential $\ell_\infty$ covering number $\mathcal{N}_\infty(\mathcal{F} \circ \mathbf{x}, \alpha, T)$ be the size of the smallest such cover. The sequential $\ell_\infty$ entropy of $\mathcal{F}$ at scale $\alpha$ and depth $T$ is defined as the logarithm of the worst-case sequential covering number: $\mathcal{H}_\infty(\mathcal{F}, \alpha, T) := \sup_{\mathbf{x}} \log \mathcal{N}_\infty(\mathcal{F} \circ \mathbf{x}, \alpha, T)$.

**Definition 3.5 (Global sequential $\ell_\infty$ cover and entropy)** Given a hypothesis class $\mathcal{F} \subseteq [0,1]^{\mathcal{X}}$, we say a collection of mappings $\mathcal{G}_\alpha \subseteq [0,1]^{\mathcal{X}*}$ is a *global* sequential cover of $\mathcal{F}$ at scale $\alpha > 0$ and depth $T$ if for any $f \in \mathcal{F}, x_{1:T} \in \mathcal{X}^T$, there exists some $g \in \mathcal{G}_\alpha$ such that

$$|f(x_t) - g(x_{1:t})| \leq \alpha, \forall t \in [T].$$

Let the *global* sequential $\ell_\infty$ covering number $\mathcal{N}_G(\mathcal{F}, \alpha, T)$ be the size of the smallest such cover. The *global* sequential $\ell_\infty$ entropy of $\mathcal{F}$ at scale $\alpha$ and depth $T$ is defined as

$$\mathcal{H}_G(\mathcal{F}, \alpha, T) := \log \mathcal{N}_G(\mathcal{F}, \alpha, T).$$

**Proposition 3.6 ([BFR20; WHGS23])** *For any $\mathcal{F} \subseteq [0,1]^{\mathcal{X}}$ and horizon $T$,*

$$\mathcal{R}_T(\mathcal{F}) \leq \min \left\{ \underbrace{\inf_{\alpha>0} \{4T\alpha + c\mathcal{H}_\infty(\mathcal{F}, \alpha, T)\}}_{[\text{BFR20}]}, \underbrace{\inf_{\alpha>0} \{T\log(1 + 2\alpha) + \mathcal{H}_G(\mathcal{F}, \alpha, T)\}}_{[\text{WHGS23}]} \right\},$$

*where $c = \frac{2 - \log(2)}{\log(3) - \log(2)} \in (3, 4)$.*

It is easy to show that $\mathcal{H}_\infty(\mathcal{F}, \alpha, T) \leq \mathcal{H}_G(\mathcal{F}, \alpha, T)$, but, in general, the two bounds in Proposition 3.6 are incomparable due to constants and different dependence on $\alpha$ (more discussions on these bounds are deferred to Appendix C). Starting from the contextual Shtarkov sum, we are able to derive a bound that combines the best of these two bounds:

**Theorem 3.7 (Main result: sequential entropy bound)** *For any $\mathcal{F} \subseteq [0,1]^{\mathcal{X}}$ and horizon $T$,*

$$\mathcal{R}_T(\mathcal{F}) \leq \inf_{\alpha>0} \left\{ T\log(1 + 2\alpha) + \mathcal{H}_\infty(\mathcal{F}, \alpha, T) \right\}.$$

**Proof** For any scale $\alpha > 0$ and depth-$T$ context tree $\mathbf{x}$, let $V_{\mathbf{x},\alpha}$ be a sequential cover of $\mathcal{F} \circ \mathbf{x}$ at scale $\alpha$ with size $\mathcal{N}_\infty(\mathcal{F} \circ \mathbf{x}, \alpha, T)$. We can always assume $V_{\mathbf{x},\alpha}$ to be $[0, 1]$-valued without loss of generality because otherwise we can just truncate it without violating its coverage guarantee. Define the smoothed covering set $\tilde{V}_{\mathbf{x},\alpha} = \left\{ \tilde{v} : \forall t \in [T], \tilde{v}_t(\cdot) = \frac{v_t(\cdot) + \alpha}{1 + 2\alpha}, v \in V_{\mathbf{x},\alpha} \right\}$, inspired by [BFR23; WHGS23]. Then for any $f \in \mathcal{F}, \mathbf{y} \in \mathcal{Y}^T$, there exists some $v \in V_{\mathbf{x},\alpha}$ such that $|f(\mathbf{x}_t(\mathbf{y})) - v_t(\mathbf{y})| \leq \alpha, \forall t \in [T]$ and hence $\tilde{v}$ satisfies

$$\frac{f(\mathbf{x}_t(\mathbf{y}))}{\tilde{v}_t(\mathbf{y})} \leq 1 + 2\alpha, \quad \frac{1 - f(\mathbf{x}_t(\mathbf{y}))}{1 - \tilde{v}_t(\mathbf{y})} \leq 1 + 2\alpha.$$

Hence

$$P_f(\mathbf{y}|\mathbf{x}(\mathbf{y})) = \prod_{t=1}^T f(\mathbf{x}_t(\mathbf{y}))^{y_t} (1 - f(\mathbf{x}_t(\mathbf{y})))^{1-y_t} \leq (1 + 2\alpha)^T \prod_{t=1}^T \tilde{v}_t(\mathbf{y})^{y_t} (1 - \tilde{v}_t(\mathbf{y}))^{1-y_t},$$

and

$$\sum_{\mathbf{y}} \sup_{f \in \mathcal{F}} P_f(\mathbf{y}|\mathbf{x}(\mathbf{y})) \leq (1 + 2\alpha)^T \sum_{\mathbf{y}} \sup_{\tilde{v} \in \tilde{V}_{\mathbf{x},\alpha}} \prod_{t=1}^T \tilde{v}_t(\mathbf{y})^{y_t} (1 - \tilde{v}_t(\mathbf{y}))^{1-y_t}$$

$$\leq (1 + 2\alpha)^T \sum_{\tilde{v} \in \tilde{V}_{\mathbf{x},\alpha}} \sum_{\mathbf{y}} \prod_{t=1}^T \tilde{v}_t(\mathbf{y})^{y_t} (1 - \tilde{v}_t(\mathbf{y}))^{1-y_t} = (1 + 2\alpha)^T |\tilde{V}_{\mathbf{x},\alpha}|,$$

where the last equality follows from Lemma D.1, treating $\tilde{v}$ as sequential experts. Finally,

$$\mathcal{R}_T(\mathcal{F}) = \sup_{\mathbf{x}} \log \left( \sum_{\mathbf{y}} \sup_{f \in \mathcal{F}} P_f(\mathbf{y}|\mathbf{x}(\mathbf{y})) \right)$$

$$\leq \sup_{\mathbf{x}} \log \left( (1 + 2\alpha)^T |\tilde{V}_{\mathbf{x},\alpha}| \right) = T \log(1 + 2\alpha) + \mathcal{H}_\infty(\mathcal{F}, \alpha, T).$$

Since our choice of $\alpha$ is arbitrary, the result follows. ∎

## 3.2 The inadequacy of sequential $\ell_\infty$ covering number

We conclude this section with a discussion on the suboptimality of regret bounds based on sequential covering numbers as in Proposition 3.6 and Theorem 3.7. Let us consider the binary label setting and the following hypothesis classes over the unit Hilbert ball $\mathcal{X} = \mathbb{B}_2$:

$$\mathcal{F}^{\text{Lin}} := \left\{ x \mapsto \frac{\langle w, x \rangle + 1}{2} : w \in \mathbb{B}_2 \right\}, \quad \mathcal{F}^{\text{AbsLin}} := \{ x \mapsto |\langle w, x \rangle| : w \in \mathbb{B}_2 \}. \tag{1}$$

We can see that the sequential $\ell_\infty$ covering numbers of $\mathcal{F}^{\text{Lin}}$ and $\mathcal{F}^{\text{AbsLin}}$ are of the same order for all scales, thus the aforementioned results will yield the same regret bound for these two classes. However, we have $\mathcal{R}_T(\mathcal{F}^{\text{Lin}}) = \tilde{O}(\sqrt{T})$ while $\mathcal{R}_T(\mathcal{F}^{\text{AbsLin}}) = \tilde{\Theta}(T^{2/3})$ [RS15; WHGS23], which implies that the sequential $\ell_\infty$ covering number, in its current form within the regret bound, cannot characterize the minimax regret.

It is worth mentioning that an $\tilde{\Omega}(\sqrt{T})$ lower bound on $\mathcal{R}_T(\mathcal{F}^{\text{Lin}})$ is achievable, via an $\tilde{\Omega}(1/\alpha^2)$ lower bound on the sequential fat-shattering dimension $\text{sfat}_\alpha(\mathcal{F}^{\text{Lin}})$ combined with Proposition 2 in [WHGS23]. The same lower bound also holds in the finite-dimensional case, where $\mathbb{B}_2$ is a unit $d$-dimensional Euclidean ball with $d \geq \sqrt{T}$ [WHGS23, Footnote 6]. Our proof (Appendix A.7) of the next result works in both the infinite and finite dimensional (with $d \geq T$) cases.

**Lemma 3.8** ($\Omega(\sqrt{T})$ **lower bound for the linear class** $\mathcal{F}^{\text{Lin}}$) *For $\mathcal{F}^{\text{Lin}}$ defined as in Eq. (1) with $\mathbb{B}_2$ being the unit Hilbert ball or the unit $d$-dimensional Euclidean ball with $d \geq T$, then*

$$\mathcal{R}_T(\mathcal{F}^{\text{Lin}}) \geq \mathcal{R}_T^{\text{FD}}(\mathcal{F}^{\text{Lin}}) \geq \sqrt{T}/4.$$

The proof of Lemma 3.8 is based on lower bounding the conditional Shtarkov sums (and hence the contextual Shtarkov sums) of $\mathcal{F}^{\text{Lin}}$. From Theorem 3.2 we know that the $\tilde{O}(\sqrt{T})$ upper bound holds for the log of contextual Shtarkov sums as well but we do not have a direct proof of this fact so far.

---

**Algorithm 1** Contextual Normalized Maximum Likelihood (cNML)

---

**Input:** Hypothesis class $\mathcal{F}$, horizon $T$

**For** $t = 1, 2, ..., T$ **do**

      1. Observe context $x_t \in \mathcal{X}$

      2. If $\sup_{f \in \mathcal{F}} P_f(y_{1:t-1}|x_{1:t-1}) > 0$, predict $\hat{p}_t \in \Delta(\mathcal{Y})$ with

$$\hat{p}_t(y) = \frac{\sup_{\mathbf{x}} S_T^{x_{1:t},(y_{1:t-1},y)}(\mathcal{F}|\mathbf{x})}{\sum_{y' \in \mathcal{Y}} \sup_{\mathbf{x}} S_T^{x_{1:t},(y_{1:t-1},y')}(\mathcal{F}|\mathbf{x})}, \forall y \in \mathcal{Y}, \tag{2}$$

      and otherwise set $\hat{p}_t$ to be an arbitrary member of $\Delta^+(\mathcal{Y})$

      3. Receive label $y_t \in \mathcal{Y}$

**End for**

---

## 4 Contextual NML, the minimax optimal algorithm

So far we have settled the minimax regret of sequential probability assignment in a nonconstructive way. Now we switch to the algorithmic lens to study the optimal strategy that achieves the minimax regret. Remarkably, we show that the minimax optimal algorithm can be described by a data-dependent variant of the contextual Shtarkov sum, which is named contextual Shtarkov sum *with prefix*.

**Definition 4.1 (Contextual Shtarkov sum with prefix)** Given sequences $x_{1:t} \in \mathcal{X}^t, y_{1:t} \in \mathcal{Y}^t, t \in [T]$ and a context tree $\mathbf{x}$ of depth $T - t$, the contextual Shtarkov sum $S_T^{x_{1:t},y_{1:t}}(\mathcal{F}|\mathbf{x})$ of $\mathcal{F}$ on $\mathbf{x}$ with prefix $x_{1:t}, y_{1:t}$ is defined as

$$S_T^{x_{1:t},y_{1:t}}(\mathcal{F}|\mathbf{x}) = \sum_{\mathbf{y} \in \mathcal{Y}^{T-t}} \sup_{f \in \mathcal{F}} P_f(y_{1:t}, \mathbf{y}|x_{1:t}, \mathbf{x}(\mathbf{y})).$$

Now we present our prediction strategy, *contextual normalized maximum likelihood* (cNML), which is summarized in Algorithm 1. In each round $t$, with $x_{1:t}, y_{1:t-1}$ as past observations, the learner first checks whether $\sup_{f \in \mathcal{F}} P_f(y_{1:t-1}|x_{1:t-1}) > 0$ since if that is not the case and $\sup_{f \in \mathcal{F}} P_f(y_{1:t-1}|x_{1:t-1}) = 0$, the cumulative losses of all experts in $\mathcal{F}$ have already blown up to $+\infty$ and the learner only needs to predict any $\hat{p} \in \Delta^+(\mathcal{Y})$ in all remaining rounds. On the other hand, if $\sup_{f \in \mathcal{F}} P_f(y_{1:t-1}|x_{1:t-1}) > 0$, then

$$\max_{y \in \mathcal{Y}} \sup_{\mathbf{x}} S_T^{x_{1:t},(y_{1:t-1},y)}(\mathcal{F}|\mathbf{x}) > 0$$

and the $\hat{p}_t$ given by Eq. (2) is indeed a valid member of $\Delta(\mathcal{Y})$ (shown in Appendix B) and is used as the learner's prediction. The following theorem shows that cNML is the minimax optimal algorithm, with proof deferred to Appendix B.

**Theorem 4.2 (Main result: optimal algorithm)** *The contextual normalized maximum likelihood strategy (Algorithm 1) is minimax optimal.*

To see that cNML is reduced to NML in the context-free setting, it suffices to consider the case where $\sup_{f \in \mathcal{F}} P_f(y_{1:T}) > 0$ since otherwise NML will simply assign 0 probability on this sequence $y_{1:T}$ and during the actual round-wise implementation of NML, it also predicts an arbitrary element from $\Delta^+(\mathcal{Y})$ in those rounds $t$ where $\sup_f P_f(y_{1:t-1}) = 0$. Now for any $y_{1:T}$ such that $\sup_{f \in \mathcal{F}} P_f(y_{1:T}) > 0$, the prediction by cNML in each round $t$ is

$$\hat{p}_t(y) = \frac{\sum_{\mathbf{y} \in \mathcal{Y}^{T-t}} \sup_{f \in \mathcal{F}} P_f(y_{1:t-1}, y, \mathbf{y})}{\sum_{\mathbf{y}' \in \mathcal{Y}^{T-t+1}} \sup_{f \in \mathcal{F}} P_f(y_{1:t-1}, \mathbf{y}')}, \forall y \in \mathcal{Y}$$

which can be summarized into a joint density over $y_{1:T}$ by

$$\hat{p}(y_{1:T}) = \frac{\sup_{f \in \mathcal{F}} P_f(y_{1:T})}{\sum_{y'_{1:T} \in \mathcal{Y}^T} \sup_{f \in \mathcal{F}} P_f(y'_{1:T})}.$$

Recall that this is exactly the NML prediction $p_{nml}(y_{1:T})$.

**Remark 4.3 (Relaxations and efficient algorithms)** One may wonder if more efficient algorithms are available when it is not easy to compute contextual Shtarkov sums with prefix. One solution is to apply the framework of admissible relaxation in [RSS12], which provides a systematic way of constructing efficient algorithms at the cost of worse regret guarantees. Notice that the worst-case log contextual Shtarkov sums with prefix constitute a trivially "admissible relaxation" since they are the exact conditional game values.

## 5   Perspectives on contextual Shtarkov sums

In this section, we provide further insight into contextual Shtarkov sums, defined in Sections 3 and 4.

### 5.1   Contextual Shtarkov sums through martingales

We can relate our characterization of the minimax regret to the more extensively studied *sequential Rademacher complexity*, which arises in online learning problems with hypothesis class $\mathcal{F} \subseteq [0, 1]^{\mathcal{X}}$ and bounded convex losses like absolute loss. Specifically, the (conditional) sequential Rademacher complexity [RST15] is defined by

$$\mathfrak{R}_T(\mathcal{F}; \mathbf{x}) := \mathbb{E}_{\boldsymbol{\varepsilon}}\Big[ \sup_{f \in \mathcal{F}} \sum_{t=1}^{T} \varepsilon_t f(\mathbf{x}_t(\boldsymbol{\varepsilon})) \Big],$$

where $\mathbf{x}$ is a depth-$T$ binary context tree and $\boldsymbol{\varepsilon} = (\varepsilon_1, \ldots, \varepsilon_T) \in \{\pm 1\}^T$ is a sequence of i.i.d. Rademacher random variables. A notable feature of $\mathfrak{R}_T(\mathcal{F}; \mathbf{x})$ is that it is the expected supremum of the sum of a martingale differences, i.e., for any $f, \mathbb{E}[\varepsilon_t f(\mathbf{x}_t(\boldsymbol{\varepsilon}))|\varepsilon_1, \ldots, \varepsilon_{t-1}] = 0$. Likewise, $S_T(\mathcal{F}|\mathbf{x})$ also admits a martingale interpretation. To see this, let $\mathcal{F} \subseteq \{(\mathcal{X} \times \mathcal{Y})^* \times \mathcal{X} \to \Delta(\mathcal{Y})\}$ and rewrite $S_T(\mathcal{F}|\mathbf{x})$ for any context tree $\mathbf{x}$:

$$S_T(\mathcal{F}|\mathbf{x}) = \sum_{\mathbf{y} \in \mathcal{Y}^T} \sup_{f \in \mathcal{F}} P_f(\mathbf{y}|\mathbf{x}(\mathbf{y})) = \mathbb{E}_{\mathbf{y}}\Big[ \sup_{f \in \mathcal{F}} \prod_{t=1}^{T} \Big( |\mathcal{Y}| \cdot f(\mathbf{x}_{1:t}(\mathbf{y}), y_{1:t-1})(y_t) \Big) \Big],$$

where $\mathbf{y} = (y_1, \ldots, y_T)$ is a sequence of i.i.d. variables following the uniform distribution over $\mathcal{Y}$. It is easy to check that $\mathbb{E}[|\mathcal{Y}| \cdot f(\mathbf{x}_{1:t}(\mathbf{y}), y_{1:t-1})(y_t)|y_1, \ldots, y_{t-1}] = 1$, and thus

$$\Big\{ \prod_{s=1}^{t} \Big( |\mathcal{Y}| \cdot f(\mathbf{x}_{1:s}(\mathbf{y}), y_{1:s-1})(y_s) \Big) \Big\}_{t \in [T]}$$

is a martingale with respect to filtration $\mathcal{F}_t = \sigma(y_1, \ldots, y_t), t \in [T]$. It would be of independent interest to study the contextual Shtarkov sums more quantitatively by developing new tools for such product-type martingales.

### 5.2   General Shtarkov sums

We can also interpret contextual Shtarkov sums as an instance of *general Shtarkov sums*, which are defined over sub-probability measures.

**Definition 5.1 (Sub-probability measure)** A set $\mathcal{P} = \{p : \mathcal{K} \to [0, 1]\}$ is a class of sub-probability measures over a finite set $\mathcal{K}$ if

$$\sum_{k \in \mathcal{K}} p(k) \leq 1, \forall p \in \mathcal{P}.$$

Due to Lemma D.1, it is easy to see that for any hypothesis class $\mathcal{F} \subseteq \{(\mathcal{X} \times \mathcal{Y})^* \times \mathcal{X} \to \Delta(\mathcal{Y})\}$ and depth-$T$ context tree $\mathbf{x}$, they induce a class

$$\mathcal{P}_{\mathcal{F}|\mathbf{x}} := \{P_f(\cdot|\mathbf{x}(\cdot)) : f \in \mathcal{F}\}$$

that is a class of sub-probability measures over $\mathcal{Y}^T$. Moreover, for any $\mathcal{F}$, depth-$(T-t)$ context tree $\mathbf{x}$ and sequences $x_{1:t} \in \mathcal{X}^t, y_{1:t} \in \mathcal{Y}^t$, the induced

$$\mathcal{P}_{\mathcal{F}^{x_{1:t}, y_{1:t}}|\mathbf{x}} := \{P_f(y_{1:t}, \cdot|x_{1:t}, \mathbf{x}(\cdot)) : f \in \mathcal{F}\}$$

is a class of sub-probability measure over $\mathcal{Y}^{T-t}$ since

$$\sum_{\mathbf{y} \in \mathcal{Y}^{T-t}} P_f(y_{1:t}, \mathbf{y} | x_{1:t}, \mathbf{x}(\mathbf{y})) = P_f(y_{1:t} | x_{1:t}) \leq 1.$$

Next we introduce the notion of general Shtarkov sum over classes of sub-probability measures.

**Definition 5.2 (General Shtarkov sum)** Given any class $\mathcal{P}$ of sub-probability measures over $\mathcal{K}$, the general Shtarkov sum of $\mathcal{P}$ is defined as

$$S(\mathcal{P}) = \sum_{k \in \mathcal{K}} \sup_{p \in \mathcal{P}} p(k).$$

With the notion of general Shtarkov sum, it is not hard to verify that the contextual Shtarkov sums with & without prefix can be interpreted as instances of general Shtarkov sums:

**Proposition 5.3** *For any horizon $T, t \in [T]$, data sequence $x_{1:t} \in \mathcal{X}^t, y_{1:t} \in \mathcal{Y}^t$, and context trees $\mathbf{x}, \mathbf{x}'$ of depth $T, T - t$ respectively, we have*

$$S_T(\mathcal{F}|\mathbf{x}) = S(\mathcal{P}_{\mathcal{F}|\mathbf{x}}), \quad S_T^{x_{1:t}, y_{1:t}}(\mathcal{F}|\mathbf{x}') = S(\mathcal{P}_{\mathcal{F}^{x_{1:t}, y_{1:t}}|\mathbf{x}'}).$$

It would be interesting to find out other instances of general Shtarkov sums that capture the complexities of other online learning problems with log loss.

## 6  Discussions

In this paper, we characterize the minimax regret and the optimal prediction strategy for sequential probability assignment, generalizing the classical results in the context-free setting. Moreover, our results are general enough to subsume the setting of multiary labels and sequential hypothesis classes, which has not been sufficiently explored before. Remarkably, our characterization holds for arbitrary hypothesis classes that may not admit the regularity assumptions implicitly required by prior works (e.g. [RST15; BFR20]).

For future works, it would be interesting to study the minimax regret of specific classes more quantitatively using our contextual Shtarkov sums. It is also intriguing to consider the setting of infinite labels. Although most of our arguments would go through under sufficient regularity conditions, a more systematic study is needed. On the practical side, it is important to develop algorithms that are more computationally efficient than cNML and with provable guarantees.

## Acknowledgements

We are grateful to Changlong Wu for telling us about Proposition C.2 and to Zeyu Jia for insights leading to Lemma 3.8. We would also like to thank Blair Bilodeau and Sasha Voitovych for helpful discussions and comments on earlier drafts of this work. ZL is supported by the Vector Research Grant at the Vector Institute. IA is supported by the Vatat Scholarship from the Israeli Council for Higher Education. DMR is supported by an NSERC Discovery Grant and funding through his Canada CIFAR AI Chair at the Vector Institute.

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

# A Proofs for Section 3

**Notations.** When the context and label sequences $x_{1:T}, y_{1:T}$ are clear from the context, we may use $f_t$ to denote the probability vector $f(x_{1:t}, y_{1:t-1}) \in \Delta(\mathcal{Y})$ produced by hypothesis $f$ at time $t$ for notational convenience. We also adopt the notation for repeated operators in [RST15; BFR20], denoting $\mathrm{Opt}_1 \cdots \mathrm{Opt}_T[\cdots]$ by $\langle\!\langle \mathrm{Opt}_t \rangle\!\rangle_{t=1}^{T} [\cdots]$. For any discrete distribution $P$ and discrete random variables $X, Y$, let $H(P)$ be the entropy of $P$ and $H(X|Y)$ be the conditional entropy of $X$ given $Y$.

## A.1 Proof overview of Theorem 3.2

Before presenting the proof of Theorem 3.2 in full details in Appendices A.2 to A.4, we give a high-level overview here.

The proof starts from swapping the pairs of inf and sup (after randomizing the labels revealed by the nature) in the extensive formulation of $\mathcal{R}_T(\mathcal{F})$ to move to the *dual game*, where the learner predicts *after* seeing the action of the nature. Trivially the value of this swapped game is a lower bound for $\mathcal{R}_T(\mathcal{F})$, and after rearranging we get that

$$\textit{the value of the swapped game} = \sup_{\mathbf{x}, \mathbf{p}} \mathbb{E}_{\mathbf{y} \sim \mathbf{p}}[\mathcal{R}_T(\mathcal{F}; \mathbf{p}(\mathbf{y}), \mathbf{x}(\mathbf{y}), \mathbf{y})] \leq \mathcal{R}_T(\mathcal{F}),$$

where the supremum is taken over all context trees $\mathbf{x}$ and probabilistic trees $\mathbf{p}$, of depth $T$. Also $\mathbb{E}_{\mathbf{y} \sim \mathbf{p}}$ means the nested conditional expectations $\mathbb{E}_{y_1 \sim \mathbf{p}_1(\mathbf{y})} \mathbb{E}_{y_2 \sim \mathbf{p}_2(\mathbf{y})} \cdots \mathbb{E}_{y_T \sim \mathbf{p}_T(\mathbf{y})}$.

Similar to the proof of Lemma 6 in [BFR20] for the binary label setting, we apply the minimax theorem with a tweak that we devise to handle multiary labels to derive that

$$\mathcal{R}_T(\mathcal{F}) = \sup_{\mathbf{x}, \mathbf{p}} \mathbb{E}_{\mathbf{y} \sim \mathbf{p}}[\mathcal{R}_T(\mathcal{F}; \mathbf{p}(\mathbf{y}), \mathbf{x}(\mathbf{y}), \mathbf{y})] \tag{3}$$

under some mild regularity condition for $\mathcal{F}$. A key observation is that the supremum over depth-$T$ probabilistic trees $\mathbf{p}$ is equivalent to the supremum over joint distributions $P$ over $\mathcal{Y}^T$. Based on this observation and some algebra, for a fixed context tree $\mathbf{x}$, the supremum over $p$ in Eq. (3) is

$$\sup_{P \in \Delta(\mathcal{Y}^T)} H(P) + \mathbb{E}_{\mathbf{y} \sim P}\big[\sup_{f \in \mathcal{F}} \log P_f(\mathbf{y}|\mathbf{x}(\mathbf{y}))\big].$$

The value of this maximization problem can be easily computed to be

$$\log\left(\sum_{\mathbf{y}} \sup_{f \in \mathcal{F}} P_f(\mathbf{y}|\mathbf{x}(\mathbf{y}))\right) = \log S_T(\mathcal{F}|\mathbf{x}).$$

Thus,

$$\mathcal{R}_T(\mathcal{F}) = \sup_{\mathbf{x}, \mathbf{p}} \mathbb{E}_{\mathbf{y} \sim \mathbf{p}}[\mathcal{R}_T(\mathcal{F}; \mathbf{p}(\mathbf{y}), \mathbf{x}(\mathbf{y}), \mathbf{y})]$$

$$= \sup_{\mathbf{x}} \sup_{P \in \Delta(\mathcal{Y}^T)} H(P) + \mathbb{E}_{\mathbf{y} \sim P}\big[\sup_{f \in \mathcal{F}} \log P_f(\mathbf{y}|\mathbf{x}(\mathbf{y}))\big] = \sup_{\mathbf{x}} \log S_T(\mathcal{F}|\mathbf{x}).$$

However, Eq. (3) is not guaranteed when there is no assumed regularity condition for $\mathcal{F}$. To get away from this, prior works have assumed a particular hypothesis is included in $\mathcal{F}$ such that the enlarged class allows for the minimax swap [RS15; BFR20]. Nevertheless, even adding a mere hypothesis may lead to suboptimal analysis for some classes $\mathcal{F}$, say when $\mathcal{R}_T(\mathcal{F})$ is of constant order. To completely get rid of any regularity assumption and obtain a unified characterization of the minimax regret for arbitrary class $\mathcal{F}$, we provide a novel argument as follows. For an arbitrary class $\mathcal{F}$, we study a smooth truncated version of it, denoted by $\mathcal{F}^\delta$ for any level $\delta \in (0, 1/2)$, such that $\mathcal{F}^\delta$ always validates the use of the minimax theorem and hence $\mathcal{R}_T(\mathcal{F}^\delta) = \sup_{\mathbf{x}} \log S_T(\mathcal{F}^\delta|\mathbf{x})$. Then we give a series of refined analysis comparing the minimax regrets and contextual Shtarkov sums of $\mathcal{F}$ and $\mathcal{F}^\delta$ that yields

$$\mathcal{R}_T(\mathcal{F}) \leq \mathcal{R}_T(\mathcal{F}^\delta) + T \log(1 + |\mathcal{Y}|\delta) = \sup_{\mathbf{x}} \log S_T(\mathcal{F}^\delta|\mathbf{x}) + T \log(1 + |\mathcal{Y}|\delta)$$

$$\leq \log\left(\sup_{\mathbf{x}} S_T(\mathcal{F}|\mathbf{x}) + \delta \cdot C(T, |\mathcal{Y}|)\right) + T \log(1 + |\mathcal{Y}|\delta),$$

where $C(T, |\mathcal{Y}|) < \infty$ is a positive constant that only depends on $T$ and $|\mathcal{Y}|$. Sending $\delta \to 0^+$ will conclude that $\mathcal{R}_T(\mathcal{F}) \leq \sup_{\mathbf{x}} \log S_T(\mathcal{F}|\mathbf{x})$, which finishes the whole proof as we already have $\mathcal{R}_T(\mathcal{F}) \geq \sup_{\mathbf{x}} \log S_T(\mathcal{F}|\mathbf{x})$ from the start.

## A.2 Minimax swap

As standard in online learning literature, we will first move to a dual game after applying a minimax swap at each round of the game. Under mild assumptions, the value of the original game coincides with the that of the swapped game. More specifically, we have:

**Lemma A.1** *Whenever $\mathcal{F}$ satisfies that for every sequence $x_{1:T} \in \mathcal{X}^T, y_{1:T} \in \mathcal{Y}^T$,*

$$\inf_{f \in \mathcal{F}} \sum_{t=1}^{T} \ell(f(x_{1:t}, y_{1:t-1}), y_t) < \infty, \tag{4}$$

*we have that*

$$\mathcal{R}_T(\mathcal{F}) = \sup_{\mathbf{x}, \mathbf{p}} \mathbb{E}_{\mathbf{y} \sim \mathbf{p}}[\mathcal{R}_T(\mathcal{F}; \mathbf{p}(\mathbf{y}), \mathbf{x}(\mathbf{y}), \mathbf{y})], \tag{5}$$

*where the supremum is taken over all $\mathcal{X}$-valued $\mathcal{Y}$-ary trees $\mathbf{x}$ and $\Delta(\mathcal{Y})$-valued $\mathcal{Y}$-ary trees $\mathbf{p}$, of depth $T$. Also $\mathbb{E}_{\mathbf{y} \sim \mathbf{p}}$ means the nested conditional expectations $\mathbb{E}_{y_1 \sim \mathbf{p}_1(\mathbf{y})} \mathbb{E}_{y_2 \sim \mathbf{p}_2(\mathbf{y})} \cdots \mathbb{E}_{y_T \sim \mathbf{p}_T(\mathbf{y})}$.*

To deal with the unboundedness of log loss in the proof, we introduce the following truncation method inspired by [BFR23; WHGS23], generalizing the one in [BFR20] which was specific to binary labels.

**Definition A.2 (Smooth truncation)** The general smooth truncation map $\tau_\delta : \Delta(\mathcal{Y}) \to \Delta(\mathcal{Y})$ is defined such that for all $p \in \Delta(\mathcal{Y})$ and $y \in \mathcal{Y}$,

$$\tau_\delta(p)(y) = \frac{p(y) + \delta}{1 + |\mathcal{Y}|\delta},$$

given threshold $\delta \in (0, 1/2)$.

It is easy to check that $\tau_\delta(p)$ is indeed a valid member in $\Delta(\mathcal{Y})$ and $\tau_\delta(p)(y) \in [\delta/(1 + |\mathcal{Y}|\delta), (1 + \delta)/(1 + |\mathcal{Y}|\delta)]$. Moreover, it is not hard to verify that $\tau_\delta(\Delta(\mathcal{Y})) = \{p \in \Delta(\mathcal{Y}) : p(y) \in [\delta/(1 + |\mathcal{Y}|\delta), (1 + \delta)/(1 + |\mathcal{Y}|\delta)], \forall y \in \mathcal{Y}\}$. We will use $\Delta^\delta(\mathcal{Y})$ to denote this image set $\tau_\delta(\Delta(\mathcal{Y}))$.

**Proof of Lemma A.1** Fix $\delta \in (0, 1/2)$. By restricting the learner's prediction $\hat{p}_t$ to $\Delta^\delta(\mathcal{Y})$, we get an upper bound on $\mathcal{R}_T(\mathcal{F})$:

$$\mathcal{R}_T(\mathcal{F}) \leq \left\langle\!\!\left\langle \sup_{x_t} \inf_{\hat{p}_t \in \Delta^\delta(\mathcal{Y})} \sup_{y_t} \right\rangle\!\!\right\rangle_{t=1}^{T} \left[ \sum_{t=1}^{T} \ell(\hat{p}_t, y_t) - \inf_{f \in \mathcal{F}} \sum_{t=1}^{T} \ell(f_t, y_t) \right]$$

$$= \left\langle\!\!\left\langle \sup_{x_t} \inf_{\hat{p}_t \in \Delta^\delta(\mathcal{Y})} \sup_{y_t} \right\rangle\!\!\right\rangle_{t=1}^{T-1} \sup_{x_T} \inf_{\hat{p}_T \in \Delta^\delta(\mathcal{Y})} \sup_{p_T} \mathbb{E}_{y_T \sim p_T} \left[ \sum_{t=1}^{T} \ell(\hat{p}_t, y_t) - \inf_{f \in \mathcal{F}} \sum_{t=1}^{T} \ell(f_t, y_t) \right].$$

Now we can apply Sion's minimax theorem [Sio58] to the function

$$A(\hat{p}_T, p_T) = \mathbb{E}_{y_T \sim p_T} \left[ \sum_{t=1}^{T} \ell(\hat{p}_t, y_t) - \inf_{f \in \mathcal{F}} \sum_{t=1}^{T} \ell(f_t, y_t) \right]$$

to derive that

$$\inf_{\hat{p}_T \in \Delta^\delta(\mathcal{Y})} \sup_{p_T \in \Delta(\mathcal{Y})} A(\hat{p}_T, p_T) = \sup_{p_T \in \Delta(\mathcal{Y})} \inf_{\hat{p}_T \in \Delta^\delta(\mathcal{Y})} A(\hat{p}_T, p_T).$$

This is because:

1. $A(\hat{p}_T, p_T)$ is convex and continuous in $\hat{p}_T$ over the compact $\Delta^\delta(\mathcal{Y})$ and

2. $A(\hat{p}_T, p_T)$ is concave and continuous in $p_T$ over the compact $\Delta(\mathcal{Y})$, which is further due to that $A(\hat{p}_T, p_T)$ is linear in $p_T$ and is bounded given Eq. (4).

Hence

$$\mathcal{R}_T(\mathcal{F}) \le \left\langle\!\!\left\langle \sup_{x_t} \inf_{\hat{p}_t \in \Delta^\delta(\mathcal{Y})} \sup_{y_t} \right\rangle\!\!\right\rangle_{t=1}^{T-1} \sup_{x_T} \sup_{p_T} \inf_{\hat{p}_T \in \Delta^\delta(\mathcal{Y})} \mathbb{E}_{y_T \sim p_T}\Big[ \sum_{t=1}^{T} \ell(\hat{p}_t, y_t) - \inf_{f \in \mathcal{F}} \sum_{t=1}^{T} \ell(f_t, y_t) \Big]$$

$$= \left\langle\!\!\left\langle \sup_{x_t} \inf_{\hat{p}_t \in \Delta^\delta(\mathcal{Y})} \sup_{y_t} \right\rangle\!\!\right\rangle_{t=1}^{T-2} \sup_{x_{T-1}} \inf_{\hat{p}_{T-1} \in \Delta^\delta(\mathcal{Y})} \sup_{p_{T-1}} \mathbb{E}_{y_{T-1} \sim p_{T-1}}$$

$$\Big[ \sum_{t=1}^{T-1} \ell(\hat{p}_t, y_t) + \sup_{x_T} \sup_{p_T}\Big[ \inf_{\hat{p}_T \in \Delta^\delta(\mathcal{Y})} \mathbb{E}_{y_T \sim p_T} \ell(\hat{p}_T, y_T) - \mathbb{E}_{y_T \sim p_T} \inf_{f \in \mathcal{F}} \sum_{t=1}^{T} \ell(f_t, y_t) \Big] \Big].$$

Again the order of $\inf_{\hat{p}_{T-1} \in \Delta^\delta(\mathcal{Y})}$ and $\sup_{p_{T-1} \in \Delta(\mathcal{Y})}$ with respect to

$$B(\hat{p}_{T-1}, p_{T-1}) = \mathbb{E}_{y_{T-1} \sim p_{T-1}} \Big[ \sum_{t=1}^{T-1} \ell(\hat{p}_t, y_t) + \sup_{x_T} \sup_{p_T}\Big[ \inf_{\hat{p}_T \in \Delta^\delta(\mathcal{Y})} \mathbb{E}_{y_T \sim p_T} \ell(\hat{p}_T, y_T) - \mathbb{E}_{y_T \sim p_T} \inf_{f \in \mathcal{F}} \sum_{t=1}^{T} \ell(f_t, y_t) \Big] \Big]$$

can be swapped due to the same reason as above, leading to

$$\mathcal{R}_T(\mathcal{F}) \le \left\langle\!\!\left\langle \sup_{x_t} \inf_{\hat{p}_t \in \Delta^\delta(\mathcal{Y})} \sup_{y_t} \right\rangle\!\!\right\rangle_{t=1}^{T-2} \sup_{x_{T-1}} \sup_{p_{T-1}} \inf_{\hat{p}_{T-1} \in \Delta^\delta(\mathcal{Y})} \mathbb{E}_{y_{T-1} \sim p_{T-1}}$$

$$\Big[ \sum_{t=1}^{T-1} \ell(\hat{p}_t, y_t) + \sup_{x_T} \sup_{p_T}\Big[ \inf_{\hat{p}_T \in \Delta^\delta(\mathcal{Y})} \mathbb{E}_{y_T \sim p_T} \ell(\hat{p}_T, y_T) - \mathbb{E}_{y_T \sim p_T} \inf_{f \in \mathcal{F}} \sum_{t=1}^{T} \ell(f_t, y_t) \Big] \Big]$$

$$= \left\langle\!\!\left\langle \sup_{x_t} \inf_{\hat{p}_t \in \Delta^\delta(\mathcal{Y})} \sup_{y_t} \right\rangle\!\!\right\rangle_{t=1}^{T-3} \sup_{x_{T-2}} \inf_{\hat{p}_{T-2} \in \Delta^\delta(\mathcal{Y})} \sup_{p_{T-2}} \mathbb{E}_{y_{T-2} \sim p_{T-2}}$$

$$\Big\{ \sum_{t=1}^{T-2} \ell(\hat{p}_t, y_t) + \sup_{x_{T-1}} \sup_{p_{T-1}} \Big[ \inf_{\hat{p}_{T-1} \in \Delta^\delta(\mathcal{Y})} \mathbb{E}_{y_{T-1} \sim p_{T-1}} \ell(\hat{p}_{T-1}, y_{T-1})$$

$$+ \mathbb{E}_{y_{T-1} \sim p_{T-1}} \sup_{x_T} \sup_{p_T} \Big[ \inf_{\hat{p}_T \in \Delta^\delta(\mathcal{Y})} \mathbb{E}_{y_T \sim p_T} \ell(\hat{p}_T, y_T) - \mathbb{E}_{y_T \sim p_T} \inf_{f \in \mathcal{F}} \sum_{t=1}^{T} \ell(f_t, y_t) \Big] \Big] \Big\}.$$

Repeating this procedure through all $T$ rounds yields

$$\mathcal{R}_T(\mathcal{F}) \le \left\langle\!\!\left\langle \sup_{x_t} \sup_{p_t} \mathbb{E}_{y_t \sim p_t} \right\rangle\!\!\right\rangle_{t=1}^{T} \sup_{f \in \mathcal{F}} \Big[ \sum_{t=1}^{T} \inf_{\hat{p}_t \in \Delta^\delta(\mathcal{Y})} \mathbb{E}_{y_t \sim p_t}[\ell(\hat{p}_t, y_t)] - \ell(f_t, y_t) \Big].$$

By Lemma A.7, we know that we do not lose too much by restricting learner's prediction to $\Delta^\delta(\mathcal{Y})$:

$$\mathcal{R}_T(\mathcal{F}) \le \left\langle\!\!\left\langle \sup_{x_t} \sup_{p_t} \mathbb{E}_{y_t \sim p_t} \right\rangle\!\!\right\rangle_{t=1}^{T} \sup_{f \in \mathcal{F}} \Big[ \sum_{t=1}^{T} \inf_{\hat{p}_t} \mathbb{E}_{y_t \sim p_t}[\ell(\hat{p}_t, y_t)] - \ell(f_t, y_t) \Big] + |\mathcal{Y}|\delta T.$$

Sending $\delta \to 0^+$ on the RHS of the above inequality, we get

$$\mathcal{R}_T(\mathcal{F}) \le \left\langle\!\!\left\langle \sup_{x_t} \sup_{p_t} \mathbb{E}_{y_t \sim p_t} \right\rangle\!\!\right\rangle_{t=1}^{T} \sup_{f \in \mathcal{F}} \Big[ \sum_{t=1}^{T} \inf_{\hat{p}_t} \mathbb{E}_{y_t \sim p_t}[\ell(\hat{p}_t, y_t)] - \ell(f_t, y_t) \Big].$$

It is easy to see that on the RHS of the above inequality, the inner infimum over $\hat{p}_t \in \Delta(\mathcal{Y})$ is achieved at $\hat{p}_t = p_t$ due to the nature of log loss. So

$$\mathcal{R}_T(\mathcal{F}) \le \left\langle\!\!\left\langle \sup_{x_t} \sup_{p_t} \mathbb{E}_{y_t \sim p_t} \right\rangle\!\!\right\rangle_{t=1}^{T} \sup_{f \in \mathcal{F}} \Big[ \sum_{t=1}^{T} \mathbb{E}_{y_t \sim p_t}[\ell(p_t, y_t)] - \ell(f_t, y_t) \Big]$$

$$= \sup_{\mathbf{x}, \mathbf{p}} \mathbb{E}_{\mathbf{y} \sim \mathbf{p}}[\mathcal{R}_T(\mathcal{F}; \mathbf{p}(\mathbf{y}), \mathbf{x}(\mathbf{y}), \mathbf{y})],$$

where in the last equality we use the compact notation of trees to further simplify our expression and this concludes the proof. ∎

**Lemma A.3** *For any hypothesis class $\mathcal{F}$ and horizon $T$,*

$$\sup_{\mathbf{x},\mathbf{p}} \mathbb{E}_{\mathbf{y}\sim\mathbf{p}}[\mathcal{R}_T(\mathcal{F};\mathbf{p}(\mathbf{y}),\mathbf{x}(\mathbf{y}),\mathbf{y})] = \sup_{\mathbf{x}} \log S_T(\mathcal{F}|\mathbf{x}). \tag{6}$$

*It is implied that whenever $\mathcal{F}$ satisfies Eq. (5), we have*

$$\mathcal{R}_T(\mathcal{F}) = \sup_{\mathbf{x}} \log S_T(\mathcal{F}|\mathbf{x}).$$

**Proof of Lemma A.3** First we can see that the outcome sequence $y_{1:T}$ generated under any tree $\mathbf{p}$ is the same thing as $y_{1:T}$ generated by its associated joint distribution over $\mathcal{Y}^T$, and vice versa. So we can replace the supremum over trees $\mathbf{p}$ in the LHS of Eq. (6) by the supremum over joint distributions $P$ over $\mathcal{Y}^T$. Hence,

$$\sup_{\mathbf{x},\mathbf{p}} \mathbb{E}_{\mathbf{y}\sim\mathbf{p}}[\mathcal{R}_T(\mathcal{F};\mathbf{p}(\mathbf{y}),\mathbf{x}(\mathbf{y}),\mathbf{y})] = \sup_{\mathbf{x},P} \mathbb{E}_{\mathbf{y}\sim P}[\mathcal{R}_T(\mathcal{F};\mathbf{p}(\mathbf{y}),\mathbf{x}(\mathbf{y}),\mathbf{y})]$$

$$= \sup_{\mathbf{x},P} \mathbb{E}_{\mathbf{y}\sim P}\Big[\sum_{t=1}^{T}\ell(P_t,y_t) - \inf_{f\in\mathcal{F}}\sum_{t=1}^{T}\ell(f_t,y_t)\Big],$$

where $P_t$ denotes the conditional distribution $P_t(\cdot|y_{1:t-1}) \in \Delta(\mathcal{Y})$ of $y_t$ under $P$ given $y_{1:t-1}$.

Now fix the context tree $\mathbf{x}$ and distribution $P$. Then we can see that $\mathbb{E}_{\mathbf{y}\sim P}[\ell(P_t,y_t)] = H(y_t|y_{1:t-1})$. So $\mathbb{E}_{\mathbf{y}\sim P}[\sum_{t=1}^{T}\ell(P_t,y_t)] = \sum_{t=1}^{T}H(y_t|y_{1:t-1}) = H(P)$. Further notice that

$$\inf_{f\in\mathcal{F}}\sum_{t=1}^{T}\ell(f_t,y_t) = \inf_{f\in\mathcal{F}}(-\log P_f(y_{1:T}|x_{1:T})) = -\sup_{f\in\mathcal{F}}\log P_f(y_{1:T}|x_{1:T}).$$

So naturally we define the map $F_{\mathbf{x}}: \mathcal{Y}^T \to \mathbb{R}\cup\{-\infty\}$ by

$$F_{\mathbf{x}}(\mathbf{y}) = \sup_{f\in\mathcal{F}}\log P_f(\mathbf{y}|\mathbf{x}(\mathbf{y})),$$

and then we see that

$$\mathbb{E}_{\mathbf{y}\sim P}\Big[\sum_{t=1}^{T}\ell(P_t,y_t) - \inf_{f\in\mathcal{F}}\sum_{t=1}^{T}\ell((f(x_t(\mathbf{y})),y_t)\Big] = H(P) + \mathbb{E}_{\mathbf{y}\sim P}[F_{\mathbf{x}}(\mathbf{y})].$$

For any given tree $\mathbf{x}$, notice that the optimization problem

$$\sup_{P\in\Delta(\mathcal{Y}^T)} H(P) + \mathbb{E}_{\mathbf{y}\sim P}[F_{\mathbf{x}}(\mathbf{y})]$$

is actually a maximization problem in the form of $\max_{P\in\Delta(\mathcal{Y}^T)} H(P) + \langle P, v\rangle$, where $v$ is some $|\mathcal{Y}|^T-$dimensional vector. According to the conjugacy between negative entropy function and log-sum-exp function, the optimal $P^*$ is given by

$$P^*(\mathbf{y}) = \frac{\exp(F_{\mathbf{x}}(\mathbf{y}))}{\sum_{\mathbf{y}'}\exp(F_{\mathbf{x}}(\mathbf{y}'))} = \frac{\sup_{f\in\mathcal{F}}P_f(\mathbf{y}|\mathbf{x}(\mathbf{y}))}{\sum_{\mathbf{y}'}\sup_{f\in\mathcal{F}}P_f(\mathbf{y}'|\mathbf{x}(\mathbf{y}'))}, \forall \mathbf{y} \in \mathcal{Y}^T.$$

Note that the above formula for $P^*$ is also valid when $F_{\mathbf{x}}(\mathbf{y}) = -\infty$ for some $\mathbf{y}$, since $P^*$ should be supported on $\{\mathbf{y}\in\mathcal{Y}^T: F_{\mathbf{x}}(\mathbf{y}) > -\infty\}$, and $F_{\mathbf{x}}(\mathbf{y})$ cannot be $-\infty$ for all $\mathbf{y}$ due to Lemma D.1. The associated value of this maximization problem is

$$\log\left(\sum_{\mathbf{y}}\exp(F_{\mathbf{x}}(\mathbf{y}))\right) = \log\left(\sum_{\mathbf{y}}\sup_{f\in\mathcal{F}}P_f(\mathbf{y}|\mathbf{x}(\mathbf{y}))\right).$$

Therefore,

$$\sup_{\mathbf{x},\mathbf{p}} \mathbb{E}_{\mathbf{y}\sim\mathbf{p}}[\mathcal{R}_T(\mathcal{F};\mathbf{p}(\mathbf{y}),\mathbf{x}(\mathbf{y}),\mathbf{y})] = \sup_{\mathbf{x},P} \mathbb{E}_{\mathbf{y}\sim P}\Big[\sum_{t=1}^{T}\ell(P_t,y_t) - \inf_{f\in\mathcal{F}}\sum_{t=1}^{T}\ell(f_t,y_t)\Big]$$

$$= \sup_{\mathbf{x}}\sup_{P}\Big\{H(P) + \mathbb{E}_{\mathbf{y}\sim P}[F_{\mathbf{x}}(\mathbf{y})]\Big\}$$

$$= \sup_{\mathbf{x}}\log\left(\sum_{\mathbf{y}}\sup_{f\in\mathcal{F}}P_f(\mathbf{y}|\mathbf{x}(\mathbf{y}))\right)$$

$$= \sup_{\mathbf{x}}\log S_T(\mathcal{F}|\mathbf{x}).$$

$\blacksquare$

In the proof of Lemma A.1, if we do not restrict the learner's prediction and simply swap the order of inf and sup to produce an inequality at each time $t$, we will reach the following folklore result.

**Lemma A.4** *For any hypothesis class $\mathcal{F}$ and horizon $T$,*

$$\mathcal{R}_T(\mathcal{F}) \geq \sup_{\mathbf{x},\mathbf{p}} \mathbb{E}_{\mathbf{y}\sim\mathbf{p}}[\mathcal{R}_T(\mathcal{F}; \mathbf{p}(\mathbf{y}), \mathbf{x}(\mathbf{y}), \mathbf{y})]. \tag{7}$$

**Proof of Lemma A.4** To get Eq. (7), we simply need to reverse the order of sup and inf at each time in the extensive formulation of minimax regret and produce an inequality:

$$\mathcal{R}_T(\mathcal{F}) = \sup_{x_1}\inf_{\hat{p}_1}\sup_{y_1}\cdots\sup_{x_T}\inf_{\hat{p}_T}\sup_{y_T}\mathcal{R}_T(\mathcal{F}; \hat{p}_{1:T}, x_{1:T}, y_{1:T})$$

$$= \left\langle\!\!\left\langle\sup_{x_t}\inf_{\hat{p}_t}\sup_{y_t}\right\rangle\!\!\right\rangle_{t=1}^{T}\left[\sum_{t=1}^{T}\ell(\hat{p}_t, y_t) - \inf_{f\in\mathcal{F}}\sum_{t=1}^{T}\ell(f(x_{1:t}, y_{1:t-1}), y_t)\right]$$

$$= \left\langle\!\!\left\langle\sup_{x_t}\inf_{\hat{p}_t}\sup_{y_t}\right\rangle\!\!\right\rangle_{t=1}^{T-1}\sup_{x_T}\inf_{\hat{p}_T}\sup_{p_T}\mathbb{E}_{y_T\sim p_T}\left[\sum_{t=1}^{T}\ell(\hat{p}_t, y_t) - \inf_{f\in\mathcal{F}}\sum_{t=1}^{T}\ell(f_t, y_t)\right]$$

$$\geq \left\langle\!\!\left\langle\sup_{x_t}\inf_{\hat{p}_t}\sup_{y_t}\right\rangle\!\!\right\rangle_{t=1}^{T-1}\sup_{x_T}\sup_{p_T}\inf_{\hat{p}_T}\mathbb{E}_{y_T\sim p_T}\left[\sum_{t=1}^{T}\ell(\hat{p}_t, y_t) - \inf_{f\in\mathcal{F}}\sum_{t=1}^{T}\ell(f_t, y_t)\right]$$

$$= \left\langle\!\!\left\langle\sup_{x_t}\inf_{\hat{p}_t}\sup_{y_t}\right\rangle\!\!\right\rangle_{t=1}^{T-1}\left[\sum_{t=1}^{T-1}\ell(\hat{p}_t, y_t) + \sup_{x_T}\sup_{p_T}\left[\inf_{\hat{p}_T}\mathbb{E}_{y_T\sim p_T}\ell(\hat{p}_T, y_T) - \mathbb{E}_{y_T\sim p_T}\inf_{f\in\mathcal{F}}\sum_{t=1}^{T}\ell(f_t, y_t)\right]\right].$$

Iterating the argument and rearranging terms as above, we will get that

$$\mathcal{R}_T(\mathcal{F}) \geq \left\langle\!\!\left\langle\sup_{x_t}\sup_{p_t}\mathbb{E}_{y_t\sim p_t}\right\rangle\!\!\right\rangle_{t=1}^{T}\sup_{f\in\mathcal{F}}\left[\sum_{t=1}^{T}\inf_{\hat{p}_t}\mathbb{E}_{y_t\sim p_t}[\ell(\hat{p}_t, y_t)] - \ell(f_t, y_t)\right]$$

$$= \left\langle\!\!\left\langle\sup_{x_t}\sup_{p_t}\mathbb{E}_{y_t\sim p_t}\right\rangle\!\!\right\rangle_{t=1}^{T}\sup_{f\in\mathcal{F}}\left[\sum_{t=1}^{T}\mathbb{E}_{y_t\sim p_t}[\ell(p_t, y_t)] - \ell(f_t, y_t)\right]$$

$$= \sup_{\mathbf{x},\mathbf{p}}\mathbb{E}_{\mathbf{y}\sim\mathbf{p}}[\mathcal{R}_T(\mathcal{F}; \mathbf{p}(\mathbf{y}), \mathbf{x}(\mathbf{y}), \mathbf{y})].$$

∎

### A.3 Smooth truncated hypothesis class

To remove the reliance on Eq. (4), we introduce a smooth truncated version of $\mathcal{F}$ that always satisfies Eq. (4) and study its minimax regret as well as contextual Shtarkov sums, compared to those of the untruncated class $\mathcal{F}$. To be more specific, we will apply the smooth truncation map to hypotheses: for any $\delta \in (0, 1/2)$ and $f : (\mathcal{X} \times \mathcal{Y})^* \times \mathcal{X} \to \Delta(\mathcal{Y})$, we use $f^\delta$ to denote its smooth truncated counterpart $\tau_\delta \circ f$; for any hypothesis class $\mathcal{F}$, we use $\mathcal{F}^\delta$ to denote the corresponding smooth truncated class $\tau_\delta \circ \mathcal{F} = \{\tau_\delta \circ f : f \in \mathcal{F}\}$. It is easy to verify that any smooth truncated class $\mathcal{F}^\delta$ satisfies Eq. (4) and hence

$$\mathcal{R}_T(\mathcal{F}^\delta) = \sup_{\mathbf{x}}\log S_T(\mathcal{F}^\delta|\mathbf{x}).$$

Next we control the effect of truncation on the minimax regret.

**Lemma A.5** *For any $\mathcal{F}, T$ and $\delta \in (0, 1/2)$,*

$$\mathcal{R}_T(\mathcal{F}) \leq \mathcal{R}_T(\mathcal{F}^\delta) + T \cdot \log(1 + |\mathcal{Y}|\delta).$$

**Proof of Lemma A.5** Fix threshold $\delta \in (0, 1/2)$ and hypothesis $f$. By Lemma A.7, for any given sequences $x_{1:T}, y_{1:T}$, there is

$$\sum_{t=1}^{T} \ell(f^\delta(x_{1:t}, y_{1:t-1}), y_t) - \sum_{t=1}^{T} \ell(f(x_{1:t}, y_{1:t-1}), y_t) \leq T \cdot \log(1 + |\mathcal{Y}|\delta). \tag{8}$$

Then, for any sequence of predictions $\hat{p}_{1:T}$,

$$
\begin{aligned}
\mathcal{R}_T(\mathcal{F}; \hat{p}_{1:T}, x_{1:T}, y_{1:T}) &= \sum_{t=1}^{T} \ell(\hat{p}_t, y_t) - \inf_{f \in \mathcal{F}} \sum_{t=1}^{T} \ell(f_t, y_t) \\
&\leq \sum_{t=1}^{T} \ell(\hat{p}_t, y_t) - \inf_{f^\delta \in \mathcal{F}^\delta} \sum_{t=1}^{T} \ell(f_t^\delta, y_t) + T \cdot \log(1 + |\mathcal{Y}|\delta) \\
&= \mathcal{R}_T(\mathcal{F}^\delta; \hat{p}_{1:T}, x_{1:T}, y_{1:T}) + T \cdot \log(1 + |\mathcal{Y}|\delta),
\end{aligned}
$$

which concludes the proof. ∎

**Lemma A.6** *There exists a constant $M(T) < \infty$ that only depends on $T$ such that for any $f, x_{1:T} \in \mathcal{X}^T, y_{1:T} \in \mathcal{Y}^T$ and $\delta \in (0, 1/2)$,*
$$P_{f^\delta}(y_{1:T}|x_{1:T}) \leq P_f(y_{1:T}|x_{1:T}) + \delta \cdot M(T).$$

**Proof of Lemma A.6** Fix threshold $\delta \in (0, 1/2)$, hypothesis $f$ and sequences $x_{1:T}, y_{1:T}$. Then

$$
\begin{aligned}
P_{f^\delta}(y_{1:T}|x_{1:T}) = \prod_{t=1}^{T} f_t^\delta(y_t) &= \prod_t \left( \frac{f_t(y_t) + \delta}{1 + |\mathcal{Y}|\delta} \right) \\
&\leq \prod_t (f_t(y_t) + \delta) \\
&= \prod_t f_t(y_t) + \delta \cdot \sum_t \prod_{t' \neq t} f_{t'}(y_{t'}) + \cdots + \delta^T \\
&\leq \prod_t f_t(y_t) + \delta \cdot M(T) \\
&= P_f(y_{1:T}|y_{1:T}) + \delta \cdot M(T),
\end{aligned}
$$

where we can set $M(T) = T + \binom{T}{2} + \binom{T}{3} + \cdots + \binom{T}{T}$ since $f_t(y_t)$'s are bounded by 1. ∎

### A.4 Putting together

Now we are fully prepared to finish the proof of Theorem 3.2, our main result in Section 3.

**Proof of Theorem 3.2** By Lemma A.6, we have that for any context tree $\mathbf{x}$ of depth $T$,

$$\sum_{\mathbf{y} \in \mathcal{Y}^T} \sup_{f^\delta \in \mathcal{F}^\delta} P_{f^\delta}(\mathbf{y}|\mathbf{x}(\mathbf{y})) \leq \sum_{\mathbf{y} \in \mathcal{Y}^T} \sup_{f \in \mathcal{F}} P_f(\mathbf{y}|\mathbf{x}(\mathbf{y})) + \delta \cdot M(T) \cdot |\mathcal{Y}|^T.$$

Thus

$$
\begin{aligned}
\mathcal{R}_T(\mathcal{F}^\delta) &= \sup_{\mathbf{x}} \log S_T(\mathcal{F}^\delta|\mathbf{x}) \\
&= \sup_{\mathbf{x}} \log \left( \sum_{\mathbf{y} \in \mathcal{Y}^T} \sup_{f^\delta \in \mathcal{F}^\delta} P_{f^\delta}(\mathbf{y}|\mathbf{x}(\mathbf{y})) \right) \\
&\leq \sup_{\mathbf{x}} \log \left( \sum_{\mathbf{y} \in \mathcal{Y}^T} \sup_{f \in \mathcal{F}} P_f(\mathbf{y}|\mathbf{x}(\mathbf{y})) + \delta \cdot M(T) \cdot |\mathcal{Y}|^T \right) \\
&= \log \left( \sup_{\mathbf{x}} \sum_{\mathbf{y} \in \mathcal{Y}^T} \sup_{f \in \mathcal{F}} P_f(\mathbf{y}|\mathbf{x}(\mathbf{y})) + \delta \cdot M(T) \cdot |\mathcal{Y}|^T \right).
\end{aligned}
$$

Together with Lemma A.5, we get that for any $\delta \in (0, 1/2)$,

$$\mathcal{R}_T(\mathcal{F}) \leq \log\left(\sup_{\mathbf{x}} \sum_{\mathbf{y} \in \mathcal{Y}^T} \sup_{f \in \mathcal{F}} P_f(\mathbf{y}|\mathbf{x}(\mathbf{y})) + \delta \cdot M(T) \cdot |\mathcal{Y}|^T\right) + T \cdot \log(1 + |\mathcal{Y}|\delta). \quad (9)$$

After sending $\delta \to 0^+$ on the RHS of Eq. (9),

$$\mathcal{R}_T(\mathcal{F}) \leq \log\left(\sup_{\mathbf{x}} \sum_{\mathbf{y} \in \mathcal{Y}^T} \sup_{f \in \mathcal{F}} P_f(\mathbf{y}|\mathbf{x}(\mathbf{y}))\right) = \sup_{\mathbf{x}} \log S_T(\mathcal{F}|\mathbf{x}).$$

Recall that we have $\mathcal{R}_T(\mathcal{F}) \geq \sup_{\mathbf{x}} \log S_T(\mathcal{F}|\mathbf{x})$ from Lemma A.4 and Lemma A.3. So finally,

$$\mathcal{R}_T(\mathcal{F}) = \sup_{\mathbf{x}} \log S_T(\mathcal{F}|\mathbf{x}).$$

∎

## A.5  Additional proofs

**Lemma A.7** *For any $p \in \Delta(\mathcal{Y})$ and $\delta \in (0, 1/2)$,*

$$\ell(\tau_\delta(p), y) \leq \ell(p, y) + \log(1 + |\mathcal{Y}|\delta) \leq \ell(p, y) + |\mathcal{Y}|\delta, \forall y \in \mathcal{Y}.$$

**Proof of Lemma A.7**  By direct computation, for any $y \in \mathcal{Y}$,

$$\begin{aligned}
\ell(\tau_\delta(p), y) - \ell(p, y) &= \log\left(\frac{p(y)}{p(y) + \delta} \cdot (1 + |\mathcal{Y}|\delta)\right) \\
&\leq \log(1 + |\mathcal{Y}|\delta) \\
&\leq |\mathcal{Y}|\delta.
\end{aligned}$$

∎

## A.6  Proof of Proposition 3.3

Starting from Theorem 3.2 that $\mathcal{R}_T(\mathcal{F}) = \sup_{\mathbf{x}} \log S_T(\mathcal{F}|\mathbf{x})$, we have

$$\begin{aligned}
\mathcal{R}_T(\mathcal{F}) &= \sup_{\mathbf{x}} \log\left(\sum_{\mathbf{y}} \sup_{f \in \mathcal{F}} P_f(\mathbf{y}|\mathbf{x}(\mathbf{y}))\right) \\
&\leq \sup_{\mathbf{x}} \log\left(\sum_{\mathbf{y}} \sum_{f \in \mathcal{F}} P_f(\mathbf{y}|\mathbf{x}(\mathbf{y}))\right) \\
&= \sup_{\mathbf{x}} \log\left(\sum_{f \in \mathcal{F}} \sum_{\mathbf{y}} P_f(\mathbf{y}|\mathbf{x}(\mathbf{y}))\right) = \log|\mathcal{F}|,
\end{aligned}$$

where the last equality is due to Lemma D.1.

## A.7  Proof of Lemma 3.8

It suffices to show that $\mathcal{R}_T^{\mathrm{FD}}(\mathcal{F}^{\mathrm{Lin}}) \geq \sqrt{T}/4$. In particular, we only need to find some context sequence $x_{1:T}$ such that $\log S_T(\mathcal{F}^{\mathrm{Lin}}|x_{1:T}) \geq \sqrt{T}/4$ due to Proposition 2.5. Here we pick $x_{1:T}$ such that $x_t$ are unit vectors and $x_t \perp x_{t'}$ whenever $t \neq t'$. Such sequence exists because the

dimension of $\mathbb{B}_2$ is no smaller than $T$. In this way, for each possible label sequence $y_{1:T} \in \{0,1\}^T$, we can see that the $f_w \in \mathcal{F}^{\mathrm{Lin}}$ that is indexed by

$$w = \sum_{t=1}^{T} \frac{2y_t - 1}{\sqrt{T}} x_t$$

achieves likelihood $P_{f_w}(y_{1:T}|x_{1:T}) = (\frac{1+1/\sqrt{T}}{2})^T$. Therefore,

$$S_T(\mathcal{F}^{\mathrm{Lin}}|x_{1:T}) = \sum_{y_{1:T} \in \{0,1\}^T} \sup_{f \in \mathcal{F}^{\mathrm{Lin}}} P_f(y_{1:T}|x_{1:T}) \geq \sum_{y_{1:T} \in \{0,1\}^T} \left( \frac{1+1/\sqrt{T}}{2} \right)^T = \left( 1 + 1/\sqrt{T} \right)^T,$$

which implies that $\mathcal{R}_T^{\mathrm{FD}}(\mathcal{F}^{\mathrm{Lin}}) = \log S_T(\mathcal{F}^{\mathrm{Lin}}|x_{1:T}) \geq T \log(1 + 1/\sqrt{T}) \geq \sqrt{T}/4$ for all $T \geq 1$.

# B  Proofs for Section 4

**Notations.**  Again we may use $f_t$ to denote the probability vector $f(x_{1:t}, y_{1:t-1}) \in \Delta(\mathcal{Y})$ produced by hypothesis $f$ at time $t$ when the context and label sequences $x_{1:T}, y_{1:T}$ are clear from the context. For a context tree $\mathbf{x}$ of depth $T-t$ and a path $\mathbf{y} \in \mathcal{Y}^{T-t}$, we re-index $\mathbf{x}(\mathbf{y})$ as $(\mathbf{x}_{t+1}(\mathbf{y}), \ldots, \mathbf{x}_T(\mathbf{y}))$ whenever it takes the last $T - t$ entries of the entire context sequence. And we do the same for the probabilistic tree $\mathbf{p}$ as well. That is, whenever $\mathbf{y} = (y_{t+1}, \ldots, y_T) \in \mathcal{Y}^{T-t}$ takes the last $T - t$ entries of the whole label sequence and $\mathbf{y} \sim \mathbf{p}$, then we will denote this label generating process by $y_{t+1} \sim \mathbf{p}_{t+1}(\mathbf{y}), \ldots, y_T \sim \mathbf{p}_T(\mathbf{y})$.

## B.1  Proof of Theorem 4.2

Recall that the minimax regret is

$$\mathcal{R}_T(\mathcal{F}) = \left\lVert \sup_{x_t} \inf_{\hat{p}_t} \sup_{y_t} \right\rVert_{t=1}^{T} \left[ \sum_{t=1}^{T} \ell(\hat{p}_t, y_t) - \inf_{f \in \mathcal{F}} \sum_{t=1}^{T} \ell(f(x_{1:t}, y_{1:t-1}), y_t) \right].$$

Through this extensive form of the minimax regret, we know that given $x_{1:t}, y_{1:t-1}$, the minimax prediction $\hat{p}_t^*$ at round $t$ is the one that minimizes the following expression over all $\hat{p}_t \in \Delta(\mathcal{Y})$:

$$\sup_{y_t} \left\lVert \sup_{x_s} \inf_{\hat{p}_s} \sup_{y_s} \right\rVert_{s=t+1}^{T} \left[ \sum_{s=t}^{T} \ell(\hat{p}_s, y_s) - \inf_{f \in \mathcal{F}} \sum_{s=1}^{T} \ell(f(x_{1:s}, y_{1:s-1}), y_s) \right]. \tag{10}$$

Define

$$G(\mathcal{F}, x_{1:t}, y_{1:t}) = \left\lVert \sup_{x_s} \inf_{\hat{p}_s} \sup_{y_s} \right\rVert_{s=t+1}^{T} \left[ \sum_{s=t+1}^{T} \ell(\hat{p}_s, y_s) - \inf_{f \in \mathcal{F}} \sum_{s=1}^{T} \ell(f(x_{1:s}, y_{1:s-1}), y_s) \right],$$

and now

$$\hat{p}_t^* = \operatorname*{argmin}_{\hat{p}_t \in \Delta(\mathcal{Y})} \sup_{y_t} \left\{ \ell(\hat{p}_t, y_t) + G(\mathcal{F}, x_{1:t}, y_{1:t}) \right\}.$$

The crux of the proof is to show the following:

**Lemma B.1** *For any hypothesis class $\mathcal{F}$ and sequences $x_{1:t} \in \mathcal{X}^t, y_{1:t} \in \mathcal{Y}^t$,*
$$G(\mathcal{F}, x_{1:t}, y_{1:t}) = \sup_{\mathbf{x}} \log S_T^{x_{1:t}, y_{1:t}}(\mathcal{F}|\mathbf{x}).$$

The proof of Lemma B.1 is done by essentially following the same strategy in Appendix A since $G(\mathcal{F}, x_{1:t}, y_{1:t})$ admits a similar extensive form with the minimax regret $\mathcal{R}_T(\mathcal{F})$. For completeness we provide its proof in Appendix B.2. Given Lemma B.1, we have

$$\hat{p}_t^* = \operatorname*{argmin}_{\hat{p}_t \in \Delta(\mathcal{Y})} \sup_{y_t} \left\{ \ell(\hat{p}_t, y_t) + \sup_{\mathbf{x}} \log S_T^{x_{1:t}, y_{1:t}}(\mathcal{F}|\mathbf{x}) \right\}$$

$$= \operatorname*{argmin}_{\hat{p}_t \in \Delta(\mathcal{Y})} \sup_{y_t} \log \left( \frac{\sup_{\mathbf{x}} S_T^{x_{1:t}, y_{1:t}}(\mathcal{F}|\mathbf{x})}{\hat{p}_t(y_t)} \right).$$

We apply the following result to solve the above program:

**Lemma B.2** *[MG22, Lemma 15] Let $g : \mathcal{Y} \to [0, +\infty]$ be a measurable function such that $\int_{\mathcal{Y}} g(y)d\mu \in (0, +\infty)$. Then,*

$$\inf_{p} \sup_{y \in \mathcal{Y}} \log \frac{g(y)}{p(y)} = \log\Big(\int_{\mathcal{Y}} g(y)\mu(dy)\Big), \tag{11}$$

*where the infimum in Eq. (11) spans over all probability densities $p : \mathcal{Y} \to [0, +\infty)$ with respect to $\mu$, and the infimum is reached at*

$$p^* = \frac{g}{\int_{\mathcal{Y}} g(y)d\mu}.$$

Letting $g(y) = \sup_{\mathbf{x}} S_T^{x_{1:t},(y_{1:t-1},y)}(\mathcal{F}|\mathbf{x}) \in [0, 1]$ and $\mu$ be the counting measure on the finite space $\mathcal{Y}$, we can apply Lemma B.2 whenever not all $g(y)$'s are 0. In this case, we solve that

$$\hat{p}_t^*(y) = \frac{g(y)}{\sum_{y' \in \mathcal{Y}} g(y')} = \frac{\sup_{\mathbf{x}} S_T^{x_{1:t},(y_{1:t-1},y)}(\mathcal{F}|\mathbf{x})}{\sum_{y' \in \mathcal{Y}} \sup_{\mathbf{x}} S_T^{x_{1:t},(y_{1:t-1},y')}(\mathcal{F}|\mathbf{x})}, \forall y \in \mathcal{Y}.$$

On the other hand, if $g(y) = 0, \forall y \in \mathcal{Y}$, then any $\hat{p}_t$ such that $\hat{p}_t(y) > 0, \forall y \in \mathcal{Y}$, is an minimax optimal prediction. Moreover, it implies that $P_f(y_{1:t-1}|x_{1:t-1}) = 0, \forall f \in \mathcal{F}$. This is because for arbitrary context tree $\mathbf{x}$,

$$0 = \sum_{y_t} \sum_{\mathbf{y} \in \mathcal{Y}^{T-t}} P_f(y_{1:t}, \mathbf{y}|x_{1:t}, \mathbf{x}(\mathbf{y}))$$

$$= \sum_{y_t} P_f(y_{1:t}|x_{1:t})$$

$$= P_f(y_{1:t-1}|x_{1:t-1}).$$

So the cumulative loss for each expert $f$ up to round $t - 1$ already blows up to $+\infty$ and the learner only needs to predict an arbitrary $\hat{p} \in \Delta^+(\mathcal{Y})$ in all remaining rounds to achieve $\mathcal{R}_T(\mathcal{F}; \hat{p}_{1:T}, x_{1:T}, y_{1:T}) = -\infty$.

Overall, we can see that the minimax optimal prediction $\hat{p}_t^* \in \Delta(\mathcal{Y})$ at round $t$ given $x_{1:t}, y_{1:t-1}$ is

$$\hat{p}_t^*(y) = \frac{\sup_{\mathbf{x}} S_T^{x_{1:t},(y_{1:t-1},y)}(\mathcal{F}|\mathbf{x})}{\sum_{y' \in \mathcal{Y}} \sup_{\mathbf{x}} S_T^{x_{1:t},(y_{1:t-1},y')}(\mathcal{F}|\mathbf{x})}, \forall y \in \mathcal{Y},$$

if there exists $y \in \mathcal{Y}$ such that $\sup_{\mathbf{x}} S_T^{x_{1:t},(y_{1:t-1},y)}(\mathcal{F}|\mathbf{x}) > 0$. Otherwise, select $\hat{p}_t^*$ to be an arbitrary element in $\Delta^+(\mathcal{Y})$ (and so do all remaining rounds).

### B.2 Auxiliary lemmas

Recall that for any hypothesis class $\mathcal{F}$ and sequences $x_{1:t} \in \mathcal{X}^t, y_{1:t} \in \mathcal{Y}^t$,

$$G(\mathcal{F}, x_{1:t}, y_{1:t}) = \left\langle\!\!\!\left\langle \sup_{x_s} \inf_{\hat{p}_s} \sup_{y_s} \right\rangle\!\!\!\right\rangle_{s=t+1}^{T} \Big[ \sum_{s=t+1}^{T} \ell(\hat{p}_s, y_s) - \inf_{f \in \mathcal{F}} \sum_{s=1}^{T} \ell(f(x_{1:s}, y_{1:s-1}), y_s) \Big]$$

$$= \left\langle\!\!\!\left\langle \sup_{x_s} \inf_{\hat{p}_s} \sup_{p_s} \mathbb{E}_{y_s \sim p_s} \right\rangle\!\!\!\right\rangle_{s=t+1}^{T} \Big[ \sum_{s=t+1}^{T} \ell(\hat{p}_s, y_s) - \inf_{f \in \mathcal{F}} \sum_{s=1}^{T} \ell(f(x_{1:s}, y_{1:s-1}), y_s) \Big].$$

To prove Lemma B.1, we need the following lemmas.

**Lemma B.3** *For any hypothesis class $\mathcal{F}$ and sequences $x_{1:t} \in \mathcal{X}^t, y_{1:t} \in \mathcal{Y}^t$,*

$$G(\mathcal{F}, x_{1:t}, y_{1:t}) \geq \sup_{\mathbf{x}, \mathbf{p}} \mathbb{E}_{\mathbf{y} \sim \mathbf{p}} \Big[ \sum_{s=t+1}^{T} \mathbb{E}_{y_s \sim \mathbf{p}_s(\mathbf{y})}[\ell(\mathbf{p}_s(\mathbf{y}), y_s)] - \inf_{f \in \mathcal{F}} \sum_{s=1}^{T} \ell(f_s, y_s) \Big]. \tag{12}$$

*And whenever for every $x_{t+1:T} \in \mathcal{X}^{T-t}, y_{t+1:T} \in \mathcal{Y}^{T-t}$, it holds*

$$\inf_{f \in \mathcal{F}} \sum_{s=1}^{T} \ell(f(x_{1:s}, y_{1:s-1}), y_s) < \infty, \tag{13}$$

*then*

$$G(\mathcal{F}, x_{1:t}, y_{1:t}) = \sup_{\mathbf{x}, \mathbf{p}} \mathbb{E}_{\mathbf{y} \sim \mathbf{p}} \Big[ \sum_{s=t+1}^{T} \mathbb{E}_{y_s \sim \mathbf{p}_s(\mathbf{y})} [\ell(\mathbf{p}_s(\mathbf{y}), y_s)] - \inf_{f \in \mathcal{F}} \sum_{s=1}^{T} \ell(f_s, y_s) \Big]. \qquad (14)$$

**Proof of Lemma B.3** First we see that similar to the proof of Lemma A.4, we can reverse every pair of sup over $p_s$ and inf over $\hat{p}_s$ in the extensive formulation of $G(\mathcal{F}, x_{1:t}, y_{1:t})$ and rearrange terms to obtain

$$G(\mathcal{F}, x_{1:t}, y_{1:t}) \geq \left\langle\!\!\left\langle \sup_{x_s} \sup_{p_s} \mathbb{E}_{y_s \sim p_s} \right\rangle\!\!\right\rangle_{s=t+1}^{T} \Big[ \sum_{s=t+1}^{T} \inf_{\hat{p}_s} \mathbb{E}_{y_s \sim p_s} [\ell(\hat{p}_s, y_s)] - \inf_{f \in \mathcal{F}} \sum_{s=1}^{T} \ell(f_s, y_s) \Big],$$

and again due to the nature of log loss,

$$G(\mathcal{F}, x_{1:t}, y_{1:t}) \geq \left\langle\!\!\left\langle \sup_{x_s} \sup_{p_s} \mathbb{E}_{y_s \sim p_s} \right\rangle\!\!\right\rangle_{s=t+1}^{T} \Big[ \sum_{s=t+1}^{T} \mathbb{E}_{y_s \sim p_s} [\ell(p_s, y_s)] - \inf_{f \in \mathcal{F}} \sum_{s=1}^{T} \ell(f_s, y_s) \Big]$$

$$= \sup_{\mathbf{x}, \mathbf{p}} \mathbb{E}_{\mathbf{y} \sim \mathbf{p}} \Big[ \sum_{s=t+1}^{T} \mathbb{E}_{y_s \sim \mathbf{p}_s(\mathbf{y})} [\ell(\mathbf{p}_s(\mathbf{y}), y_s)] - \inf_{f \in \mathcal{F}} \sum_{s=1}^{T} \ell(f_s, y_s) \Big],$$

where in the last step we compress the expression using trees (of depth $T - t$) and Eq. (12) is proved.

To show that the minimax swap is valid under Eq. (13), we follow the same strategy as in the proof of Lemma A.1 by restricting the learner's prediction $\hat{p}_s$ to $\Delta^{\delta}(\mathcal{Y})$ for any threshold $\delta \in (0, 1/2)$ which yields

$$G(\mathcal{F}, x_{1:t}, y_{1:t}) \leq \left\langle\!\!\left\langle \sup_{x_s} \sup_{p_s} \mathbb{E}_{y_s \sim p_s} \right\rangle\!\!\right\rangle_{s=t+1}^{T} \Big[ \sum_{s=t+1}^{T} \inf_{\hat{p}_s \in \Delta^{\delta}(\mathcal{Y})} \mathbb{E}_{y_s \sim p_s} [\ell(\hat{p}_s, y_s)] - \inf_{f \in \mathcal{F}} \sum_{s=1}^{T} \ell(f_s, y_s) \Big]$$

$$\leq \left\langle\!\!\left\langle \sup_{x_s} \sup_{p_s} \mathbb{E}_{y_s \sim p_s} \right\rangle\!\!\right\rangle_{s=t+1}^{T} \Big[ \sum_{s=t+1}^{T} \inf_{\hat{p}_s} \mathbb{E}_{y_s \sim p_s} [\ell(\hat{p}_s, y_s)] - \inf_{f \in \mathcal{F}} \sum_{s=1}^{T} \ell(f_s, y_s) \Big] + |\mathcal{Y}| \delta T.$$

So Eq. (14) is proved by sending $\delta \to 0^+$ on the RHS of the last inequality and the established Eq. (12). ∎

**Lemma B.4** *For any hypothesis class $\mathcal{F}$ and sequences $x_{1:t} \in \mathcal{X}^t, y_{1:t} \in \mathcal{Y}^t$,*

$$\sup_{\mathbf{x}, \mathbf{p}} \mathbb{E}_{\mathbf{y} \sim \mathbf{p}} \Big[ \sum_{s=t+1}^{T} \mathbb{E}_{y_s \sim \mathbf{p}_s(\mathbf{y})} [\ell(\mathbf{p}_s(\mathbf{y}), y_s)] - \inf_{f \in \mathcal{F}} \sum_{s=1}^{T} \ell(f_s, y_s) \Big] = \sup_{\mathbf{x}} \log S_T^{x_{1:t}, y_{1:t}}(\mathcal{F}|\mathbf{x}).$$

**Proof of Lemma B.4** The proof follows that of Lemma A.3. By replacing the probabilistic tree $\mathbf{p}$ by the joint distribution $P \in \Delta(\mathcal{Y}^{T-t})$, we get

$$\sup_{\mathbf{x}, \mathbf{p}} \mathbb{E}_{\mathbf{y} \sim \mathbf{p}} \Big[ \sum_{s=t+1}^{T} \mathbb{E}_{y_s \sim \mathbf{p}_s(\mathbf{y})} [\ell(\mathbf{p}_s(\mathbf{y}), y_s)] - \inf_{f \in \mathcal{F}} \sum_{s=1}^{T} \ell(f_s, y_s) \Big]$$

$$= \sup_{\mathbf{x}, P} \mathbb{E}_{\mathbf{y} \sim P} \Big[ \sum_{s=t+1}^{T} \ell(P_s, y_s) - \inf_{f \in \mathcal{F}} \sum_{s=1}^{T} \ell(f_s, y_s) \Big]$$

$$= \sup_{\mathbf{x}} \sup_{P \in \Delta(\mathcal{Y}^{T-t})} H(P) + \mathbb{E}_{\mathbf{y} \sim P} \Big[ \sup_{f \in \mathcal{F}} \log P_f(y_{1:t}, \mathbf{y} | x_{1:t}, \mathbf{x}(\mathbf{y})) \Big].$$

Similarly, for any fixed $\mathbf{x}$, define the map $F_{\mathbf{x}}^{x_{1:t}, y_{1:t}} : \mathcal{Y}^{T-t} \to \mathbb{R} \cup \{-\infty\}$ by

$$F_{\mathbf{x}}^{x_{1:t}, y_{1:t}}(\mathbf{y}) = \sup_{f \in \mathcal{F}} \log P_f(y_{1:t}, \mathbf{y} | x_{1:t}, \mathbf{x}(\mathbf{y})),$$

and now we solve
$$\sup_{P\in\Delta(\mathcal{Y}^{T-t})} H(P) + \mathbb{E}_{\mathbf{y}\sim P}[F_{\mathbf{x}}^{x_{1:t},y_{1:t}}(\mathbf{y})].$$
If there exists some $\mathbf{y}\in\mathcal{Y}^{T-t}$ such that $F_{\mathbf{x}}^{x_{1:t},y_{1:t}}(\mathbf{y}) > -\infty$, then the optimal $P^*$ is given by
$$P^*(\mathbf{y}) = \frac{\exp(F_{\mathbf{x}}^{x_{1:t},y_{1:t}}(\mathbf{y}))}{\sum_{\mathbf{y}'}\exp(F_{\mathbf{x}}^{x_{1:t},y_{1:t}}(\mathbf{y}'))} = \frac{\sup_{f\in\mathcal{F}} P_f(y_{1:t},\mathbf{y}|x_{1:t},\mathbf{x}(\mathbf{y}))}{\sum_{\mathbf{y}'}\sup_{f\in\mathcal{F}} P_f(y_{1:t},\mathbf{y}'|x_{1:t},\mathbf{x}(\mathbf{y}'))}, \forall \mathbf{y}\in\mathcal{Y}^{T-t},$$
and then
$$\sup_{\mathbf{x},\mathbf{p}} \mathbb{E}_{\mathbf{y}\sim\mathbf{p}}\Big[\sum_{s=t+1}^{T} \mathbb{E}_{y_s\sim\mathbf{p}_s(\mathbf{y})}[\ell(\mathbf{p}_s(\mathbf{y}),y_s)] - \inf_{f\in\mathcal{F}}\sum_{s=1}^{T}\ell(f_s,y_s)\Big]$$
$$= \sup_{\mathbf{x}} \sup_{P\in\Delta(\mathcal{Y}^{T-t})} H(P) + \mathbb{E}_{\mathbf{y}\sim P}[F_{\mathbf{x}}^{x_{1:t},y_{1:t}}(\mathbf{y})]$$
$$= \sup_{\mathbf{x}} \log\Big(\sum_{\mathbf{y}}\sup_{f\in\mathcal{F}} P_f(y_{1:t},\mathbf{y}|x_{1:t},\mathbf{x}(\mathbf{y}))\Big)$$
$$= \sup_{\mathbf{x}} \log S_T^{x_{1:t},y_{1:t}}(\mathcal{F}|\mathbf{x}).$$
However, if $F_{\mathbf{x}}^{x_{1:t},y_{1:t}}(\mathbf{y}) = -\infty$ for all $\mathbf{y}$, then it implies that for any context tree $\mathbf{x}$, path $\mathbf{y}$, and $f\in\mathcal{F}$, $P_f(y_{1:t},\mathbf{y}|x_{1:t},\mathbf{x}(\mathbf{y})) = 0$ and hence,
$$\sup_{\mathbf{x},\mathbf{p}} \mathbb{E}_{\mathbf{y}\sim\mathbf{p}}\Big[\sum_{s=t+1}^{T} \mathbb{E}_{y_s\sim\mathbf{p}_s(\mathbf{y})}[\ell(\mathbf{p}_s(\mathbf{y}),y_s)] - \inf_{f\in\mathcal{F}}\sum_{s=1}^{T}\ell(f_s,y_s)\Big]$$
$$= \sup_{\mathbf{x}} \sup_{P\in\Delta(\mathcal{Y}^{T-t})} H(P) + \mathbb{E}_{\mathbf{y}\sim P}[F_{\mathbf{x}}^{x_{1:t},y_{1:t}}(\mathbf{y})]$$
$$= -\infty$$
$$= \sup_{\mathbf{x}} \log\Big(\sum_{\mathbf{y}}\sup_{f\in\mathcal{F}} P_f(y_{1:t},\mathbf{y}|x_{1:t},\mathbf{x}(\mathbf{y}))\Big)$$
$$= \sup_{\mathbf{x}} \log S_T^{x_{1:t},y_{1:t}}(\mathcal{F}|\mathbf{x}),$$
which finishes our proof. ∎

Now we are able to prove the key result Lemma B.1.

**Proof of Lemma B.1** Fix any hypothesis class $\mathcal{F}$ and sequences $x_{1:t}\in\mathcal{X}^t, y_{1:t}\in\mathcal{Y}^t$. First we know
$$G(\mathcal{F},x_{1:t},y_{1:t}) \geq \sup_{\mathbf{x}} \log S_T^{x_{1:t},y_{1:t}}(\mathcal{F}|\mathbf{x})$$
due to Eq. (12) and Lemma B.4. For the other direction, let us fix any threshold value $\delta\in(0,1/2)$ and then
$$G(\mathcal{F},x_{1:t},y_{1:t}) \leq G(\mathcal{F}^\delta,x_{1:t},y_{1:t}) + T\cdot\log(1+|\mathcal{Y}|\delta)$$
$$= \sup_{\mathbf{x}} \log S_T^{x_{1:t},y_{1:t}}(\mathcal{F}^\delta|\mathbf{x}) + T\cdot\log(1+|\mathcal{Y}|\delta)$$
$$= \sup_{\mathbf{x}} \log\Big(\sum_{\mathbf{y}\in\mathcal{Y}^{T-t}}\sup_{f^\delta\in\mathcal{F}^\delta} P_{f^\delta}(y_{1:t},\mathbf{y}|x_{1:t},\mathbf{x}(\mathbf{y}))\Big) + T\cdot\log(1+|\mathcal{Y}|\delta)$$
$$\leq \sup_{\mathbf{x}} \log\Big(\sum_{\mathbf{y}\in\mathcal{Y}^{T-t}}\sup_{f\in\mathcal{F}} P_f(y_{1:t},\mathbf{y}|x_{1:t},\mathbf{x}(\mathbf{y})) + \delta\cdot M(T)\cdot|\mathcal{Y}|^T\Big) + T\cdot\log(1+|\mathcal{Y}|\delta)$$
$$= \log\Big(\sup_{\mathbf{x}}\sum_{\mathbf{y}\in\mathcal{Y}^{T-t}}\sup_{f\in\mathcal{F}} P_f(y_{1:t},\mathbf{y}|x_{1:t},\mathbf{x}(\mathbf{y})) + \delta\cdot M(T)\cdot|\mathcal{Y}|^T\Big) + T\cdot\log(1+|\mathcal{Y}|\delta),$$
where we have applied Lemma A.7, Lemma B.3, Lemma B.4, and Lemma A.6 accordingly. Similarly, we send $\delta\to 0^+$ on the RHS of the last inequality and get
$$G(\mathcal{F},x_{1:t},y_{1:t}) \leq \sup_{\mathbf{x}} \log\Big(\sum_{\mathbf{y}\in\mathcal{Y}^{T-t}}\sup_{f\in\mathcal{F}} P_f(y_{1:t},\mathbf{y}|x_{1:t},\mathbf{x}(\mathbf{y}))\Big) = \sup_{\mathbf{x}} \log S_T^{x_{1:t},y_{1:t}}(\mathcal{F}|\mathbf{x}),$$
which concludes the proof. ∎

## C  Additional discussions

### C.1  On the time-variant context space

In this section we generalize our analysis to the setting where the context space can evolve over time. We model time-varying context sets by a sequence of maps $\mathcal{X}_t : \mathcal{X}^{t-1} \times \mathcal{Y}^{t-1} \to 2^{\mathcal{X}}, t \in [T]$ as in [RS15; BFR20]. In each round $t$, instead of picking any context from $\mathcal{X}$, the nature is now required to only choose $x_t$ from $\mathcal{X}_t(x_{1:t-1}, y_{1:t-1}) \subseteq \mathcal{X}$. Then the minimax regret with respect to $(\mathcal{X}_t)_{t\in[T]}$ is rewritten as

$$\mathcal{R}_T(\mathcal{F}) = \left\langle\!\!\!\left\langle \sup_{x_t \in \mathcal{X}_t(x_{1:t-1}, y_{1:t-1})} \inf_{\hat{p}_t} \sup_{y_t} \right\rangle\!\!\!\right\rangle_{t=1}^{T} \mathcal{R}_T(\mathcal{F}; \hat{p}_{1:T}, x_{1:T}, y_{1:T}).$$

A context tree $\mathbf{x}$ is *consistent* with respect to $(\mathcal{X}_t)_{t\in[T]}$ if for all $t \in [T]$ and $\mathbf{y} \in \mathcal{Y}^T$, $\mathbf{x}_t(\mathbf{y}) \in \mathcal{X}_t(x_{1:t-1}, y_{1:t-1})$. Then our results in Section 3 and Section 4 can be generalized simply by replacing the supremum over all context trees (of depth $T$) by the supremum over all consistent context trees. For example, we will have

$$\mathcal{R}_T(\mathcal{F}) = \sup_{\mathbf{x} : \mathbf{x} \text{ is consistent}} \log S_T(\mathcal{F}|\mathbf{x}).$$

### C.2  On the global and non-global sequential cover

Now we go back to consider the usual setting of binary label and constant experts, i.e., $\mathcal{Y} = \{0, 1\}$ and $\mathcal{F} \subseteq [0, 1]^{\mathcal{X}}$. As mentioned in Section 3, previous works [BFR20; WHGS23] provided regret upper bounds based on $\ell_\infty$ sequential entropy. More specifically, both of their bounds are in the form of $O(\inf_{\alpha>0}\{\alpha T + \mathcal{H}(\mathcal{F}, \alpha, T)\})$, with $\mathcal{H}(\mathcal{F}, \alpha, T)$ being either the non-global entropy $\mathcal{H}_\infty(\mathcal{F}, \alpha, T)$ or the global entropy $\mathcal{H}_G(\mathcal{F}, \alpha, T)$. It is then natural to ask which one of these two bounds is tighter. It is straightforward to prove that $\mathcal{H}_\infty(\mathcal{F}, \alpha, T)$ is no larger than $\mathcal{H}_G(\mathcal{F}, \alpha, T)$. In fact, the gap between them is at most a polylog factor, as we state and prove below.[1] The proof of $\mathcal{H}_\infty(\mathcal{F}, \alpha, T) \leq \mathcal{H}_G(\mathcal{F}, \alpha, T)$ is also included for completeness. Before stating the results, we introduce the definition of sequential fat-shattering dimension.

**Definition C.1** We say an $\mathcal{X}$-valued binary tree $\mathbf{x}$ of depth $d$ is $\alpha$-shattered by a class $\mathcal{F} \subseteq [0, 1]^{\mathcal{X}}$ for some $\alpha > 0$, if there exists a $[0, 1]$-valued binary tree $\mathbf{s}$ of depth $d$ such that

$$\forall \mathbf{y} \in \{0, 1\}^d, \exists f \in \mathcal{F}, \text{ s.t. } (2y_t - 1) \cdot (f(\mathbf{x}_t(\mathbf{y})) - \mathbf{s}_t(\mathbf{y})) \geq \frac{\alpha}{2}, \forall t \in [d].$$

In this case, $\mathbf{s}$ is called the witness of the shattering. The sequential fat-shattering dimension of $\mathcal{F}$ at scale $\alpha$, denoted by $\text{sfat}_\alpha(\mathcal{F})$, is the largest $d$ such that some depth-$d$ context tree is $\alpha$-shattered by $\mathcal{F}$.

**Proposition C.2** *For any scale $\alpha > 0$, we have*

$$\mathcal{H}_\infty(\mathcal{F}, \alpha, T) \geq \min\{T, \sup_{\alpha' > \alpha} \text{sfat}_{2\alpha'}(\mathcal{F})\} \cdot \log(2).$$

*Therefore, together with $\mathcal{H}_\infty(\mathcal{F}, \alpha, T) \leq \mathcal{H}_G(\mathcal{F}, \alpha, T)$ and the folklore $\mathcal{H}_G(\mathcal{F}, \alpha, T) \leq O(\text{sfat}_\alpha(\mathcal{F}) \log(T/\alpha))$, we conclude that the regret upper bounds $O(\inf_{\alpha>0}\{\alpha T + \mathcal{H}(\mathcal{F}, \alpha, T)\}), \mathcal{H} \in \{\mathcal{H}_\infty, \mathcal{H}_G\}$, differ by at most a polylog factor.*

**Proof of Proposition C.2** Fix any $\alpha' > \alpha > 0$ and let $d_{\alpha'}$ denote $\min\{T, \text{sfat}_{2\alpha'}(\mathcal{F})\}$. Then there exists a context tree $\mathbf{x}$ and a witness tree $\mathbf{s}$, both of depth $d_{\alpha'}$, such that for any path $\mathbf{y} \in \{0, 1\}^{d_{\alpha'}}$, there exists an $f \in \mathcal{F}$ such that

$$\forall t \in [d_{\alpha'}], (2y_t - 1) \cdot (f(x_t(\mathbf{y})) - s_t(\mathbf{y})) \geq \alpha' > \alpha. \tag{15}$$

---

[1]This result and its detailed proof sketch were communicated to the first author by Changlong Wu. The argument here differs only in minor details that we introduced, perhaps unnecessarily, to arrive at a rigorous proof.

Let $V_{\mathbf{x},\alpha}$ be an arbitrary sequential $\ell_\infty$ covering of $\mathcal{F}$ on $\mathbf{x}$. Now we select a path $\mathbf{y}$ and a sequence of subsets $V_{\mathbf{x},\alpha}^{(t)} \subseteq V_{\mathbf{x},\alpha}, t \in [d_{\alpha'}]$ in the following recursive way. Define $V_{\mathbf{x},\alpha}^{(0)} = V_{\mathbf{x},\alpha}$. For each $t \in [d_{\alpha'}]$, choose $y_t \in \{0,1\}$ such that $2y_t - 1 \in \{-1,+1\}$ is the minority among all $\mathrm{sgn}(v_t(y_{1:t-1}) - s_t(y_{1:t-1})), v \in V_{\mathbf{x},\alpha}^{(t-1)}$ (ignoring those of 0's). Finally update $V_{\mathbf{x},\alpha}^{(t)} = \{v \in V_{\mathbf{x},\alpha}^{(t-1)} : \mathrm{sgn}(v_t(y_{1:t-1}) - s_t(y_{1:t-1})) = 2y_t - 1\}$.

First we argue that, if there is any time $t' \in [d_{\alpha'}]$ such that $V_{\mathbf{x},\alpha}^{(t'-1)} \neq \emptyset, V_{\mathbf{x},\alpha}^{(t')} = \emptyset$, then $V_{\mathbf{x},\alpha}$ is not a valid cover of $\mathcal{F}$ on $\mathbf{x}$. Otherwise, recall we have selected $y_1, \ldots, y_{t'-1}$. Now pick an arbitrary $y_{t'} \in \{0,1\}$. By Eq. (15) we can find some $f \in \mathcal{F}$ such that $(2y_t - 1) \cdot (f(x_t(y_{1:t-1})) - s_t(y_{1:t-1})) > \alpha, \forall t \in [t']$. Since $V_{\mathbf{x},\alpha}$ is a covering at scale $\alpha$, there is $v \in V_{\mathbf{x},\alpha}$ such that $|v_t(\mathbf{y}) - f(x_t(\mathbf{y}))| \leq \alpha, \forall t \in [t']$. This implies that $\mathrm{sgn}(f(x_t(\mathbf{y})) - s_t(\mathbf{y})) = \mathrm{sgn}(v_t(\mathbf{y}) - s_t(\mathbf{y})) = 2y_t - 1, \forall t \in [t']$. So we can always find some member of $V_{\mathbf{x},\alpha}^{(t'-1)}$ to match the minority sign of $v_{t'}(y_{1:t'-1}) - s_{t'}(y_{1:t'-1}), v \in V_{\mathbf{x},\alpha}^{(t'-1)}$, which means that $V_{\mathbf{x},\alpha}^{(t')} \neq \emptyset$ and yields a contradiction.

Now we know that $|V_{\mathbf{x},\alpha}^{(t)}| \geq 1, \forall t \in [d_{\alpha'}]$. By design $|V_{\mathbf{x},\alpha}^{(t)}| \leq |V_{\mathbf{x},\alpha}^{(t-1)}|/2, \forall t \in [d_{\alpha'}]$, so we must have $|V_{\mathbf{x},\alpha}| = |V_{\mathbf{x},\alpha}^{(0)}| \geq 2^{d_{\alpha'}}$. As the choice of covering is arbitrary, the covering number $\mathcal{N}_\infty(\mathcal{F} \circ \mathbf{x}, \alpha, d_{\alpha'})$ is also lower bounded by $2^{d_{\alpha'}}$ and hence $\mathcal{H}_\infty(\mathcal{F}, \alpha, d_{\alpha'}) \geq d_{\alpha'} \cdot \log(2)$. If $\sup_{\alpha' > \alpha} \mathrm{sfat}_{2\alpha'}(\mathcal{F}) \leq T$, then we get that

$$\mathcal{H}_\infty(\mathcal{F}, \alpha, T) \geq \sup_{\alpha' > \alpha} \mathcal{H}_\infty(\mathcal{F}, \alpha, \mathrm{sfat}_{2\alpha'}(\mathcal{F})) \geq \sup_{\alpha' > \alpha} \mathrm{sfat}_{2\alpha'}(\mathcal{F}) \cdot \log(2).$$

If there is some $\alpha' > \alpha$ such that $\mathrm{sfat}_{2\alpha'}(\mathcal{F}) \geq T$, then

$$\mathcal{H}_\infty(\mathcal{F}, \alpha, T) = \mathcal{H}_\infty(\mathcal{F}, \alpha, d_{\alpha'}) \geq T \cdot \log(2).$$

Combining these two cases together, we have

$$\mathcal{H}_\infty(\mathcal{F}, \alpha, T) \geq \min\{T, \sup_{\alpha' > \alpha} \mathrm{sfat}_{2\alpha'}(\mathcal{F})\} \cdot \log(2).$$

$\blacksquare$

**Proposition C.3** *Let $\mathcal{G}_\alpha$ be a global sequential $\alpha$-covering of $\mathcal{F}$ as defined in [WHGS23]. Then for any context tree $\mathbf{x}$, there exists a sequential cover $V_{\mathbf{x},\alpha}$ of $\mathcal{F} \circ \mathbf{x}$ at scale $\alpha$ with $|V_{\mathbf{x},\alpha}| \leq |\mathcal{G}_\alpha|$. This implies that $\mathcal{H}_\infty(\mathcal{F}, \alpha, T) \leq \log|\mathcal{G}_\alpha|$.*

**Proof of Proposition C.3** Fix arbitrary context tree $\mathbf{x}$. For any $g \in \mathcal{G}_\alpha$, define the $[0,1]$-valued tree $v^g$ by $v_t^g(\mathbf{y}) = g(x_{1:t}(\mathbf{y})), \forall t \in [T], \mathbf{y} \in \mathcal{Y}^T$. Now let $V_{\mathbf{x},\alpha} = \{v^g : g \in \mathcal{G}_\alpha\}$ and we will show that $V_{\mathbf{x},\alpha}$ is indeed a sequential cover of $\mathcal{F} \circ \mathbf{x}$ at scale $\alpha$.

For any $f \in \mathcal{F}$ and $\mathbf{y} \in \mathcal{Y}^T$, tree $\mathbf{x}$ yields a length$-T$ sequence $x_{1:T}(\mathbf{y})$ and by definition of the global sequential covering, there exists $g \in \mathcal{G}_\alpha$ such that

$$|f(x_t(\mathbf{y})) - g(x_{1:t}(\mathbf{y}))| \leq \alpha, \forall t \in [T].$$

So by our construction of $V_{\mathbf{x},\alpha}$, $v^g \in V_{\mathbf{x},\alpha}$ holds

$$|f(x_t(\mathbf{y})) - v_t^g(\mathbf{y})| = |f(x_t(\mathbf{y})) - g(x_{1:t}(\mathbf{y}))| \leq \alpha, \forall t \in [T],$$

which yields our claim after observing $|V_{\mathbf{x},\alpha}| \leq |\mathcal{G}_\alpha|$. $\blacksquare$

# D    Additional proofs

**Lemma D.1** *For any $\mathcal{X}$-valued $\mathcal{Y}$-ary context tree $\mathbf{x}$ of depth $T$, and $f : (\mathcal{X} \times \mathcal{Y})^* \times \mathcal{X} \to \Delta(\mathcal{Y})$, we have*

$$\sum_{\mathbf{y} \in \mathcal{Y}^T} P_f(\mathbf{y}|\mathbf{x}(\mathbf{y})) = 1, \tag{16}$$

*where we recall that $\mathbf{x}(\mathbf{y})$ denotes the context sequence $(\mathbf{x}_1(\mathbf{y}), \ldots, \mathbf{x}_T(\mathbf{y}))$.*

**Proof of Lemma D.1**  This is done by induction on the depth $T$. The key observation is that for any label sequence $\mathbf{y}$, $\mathbf{x}_t(\mathbf{y}) = \mathbf{x}_t(y_1, \ldots, y_{t-1})$ only depends on the first $t-1$ labels. For $T = 1$, any context tree $\mathbf{x}$ is represented by its root node $\mathbf{x}_1(\cdot) = \mathbf{x}_1 \in \mathcal{X}$ and hence

$$\sum_{y_1} P_f(y_1|\mathbf{x}_1) = \sum_{y_1} f(\mathbf{x}_1)(y_1) = 1.$$

Suppose Eq. (16) holds for all context trees $\mathbf{x}$ of depth $T \leq d$ and all sequential functions $f$. Now given any context tree $\mathbf{x} = (\mathbf{x}_1, \ldots, \mathbf{x}_{d+1})$ of depth $T = d+1$, we denote its depth $d$ subtree $(\mathbf{x}_1, \ldots, \mathbf{x}_d)$ by $\mathbf{x}_{[d]}$. Then

$$
\begin{aligned}
\sum_{\mathbf{y} \in \mathcal{Y}^{d+1}} P_f(\mathbf{y}|\mathbf{x}(\mathbf{y})) &= \sum_{y_{1:d}} \sum_{y_{d+1}} P_f(y_{1:d+1}|\mathbf{x}_1, \mathbf{x}_2(y_1), \ldots, \mathbf{x}_{d+1}(y_{1:d})) \\
&= \sum_{y_{1:d}} \sum_{y_{d+1}} P_f(y_{1:d}|\mathbf{x}_1, \ldots, \mathbf{x}_d(y_{1:d-1})) \cdot f(\mathbf{x}_1, \ldots, \mathbf{x}_{d+1}(y_{1:d}), y_{1:d})(y_{d+1}) \\
&= \sum_{y_{1:d}} P_f(y_{1:d}|\mathbf{x}_1, \ldots, \mathbf{x}_d(y_{1:d-1})) \sum_{y_{d+1}} f(\mathbf{x}_1, \ldots, \mathbf{x}_{d+1}(y_{1:d}), y_{1:d})(y_{d+1}) \\
&= \sum_{y_{1:d}} P_f(y_{1:d}|\mathbf{x}_1, \ldots, \mathbf{x}_d(y_{1:d-1})) \\
&= \sum_{\mathbf{y} \in \mathcal{Y}^d} P_f(\mathbf{y}|\mathbf{x}_{[d]}(\mathbf{y})) = 1,
\end{aligned}
$$

where the last step is due to induction. We are done. $\blacksquare$

