# OpenReview forum: "Sequential Probability Assignment with Contexts: Minimax Regret, Contextual Shtarkov Sums, and Contextual Normalized Maximum Likelihood"
_NeurIPS.cc/2024/Conference — NeurIPS 2024 poster_

### Official Review · Reviewer_qSjy · 2024-06-12

**Soundness:** 4
**Presentation:** 4
**Contribution:** 4
**Rating:** 8
**Confidence:** 5

**Summary:**

This paper studies the problem of sequential probability assignment under logarithmic loss with context and assumes that both the contexts and labels are generated adversarially. The main contribution is a complete characterization of the minimax regret via a new complexity measure named "contextual Shartkov sum." In particular, this allows the authors to recover and improve several previously known bounds with simpler proofs. The authors further demonstrate that such minimax regret can be achieved via a contextual normalized maximum likelihood predictor.

**Strengths:**

Sequential probability assignment is a fundamental problem that finds many applications, such as universal compression, portfolio optimization, interactive decision making, and even the famous next-token prediction in LLMs. This is a clear significant result, which provides the first precise minimax regret of adversarial sequential probability assignment under log-loss and provides an explicit algorithm achieving such regrets. Given the importance and foundational nature of the problem, I would expect the paper to have a broad impact on the community.

Although most of the proof ideas, such as the switching order of inf and sup, follow from prior literature, the truly novel technique that bypassed the difficulties faced by prior results is Lemma A.3. This allows the authors to characterize the minimax value directly (leveraging properties of log-loss) without resorting to the offset Rademacher complexity as in [RS15]. I also find the optimization problem in line 206 to be of independent interest. Although the paper is fairly preliminary (see point 4 in the Weakness section), it would serve as a stepping stone for important future research.

Overall, this is a significant result that would inspire follow-up research. Therefore, I recommend a "Strong Accept."

**Weaknesses:**

I do not see any significant weaknesses in the paper. However, I would like to outline a few minor remarks as follows:

1. I believe the characterization in the current paper is similar in spirit to that of [BHS23], who provides a general reduction from the smoothed adversarial to the fixed design regret that also leverages the minimax switches. (Although I believe the techniques developed in the current paper could also recover this result.)

2. I think Algorithm 1 can be fit into the relaxation-based algorithmic framework introduced by [RSS12]. (This does not mean the current algorithm is not novel.)

3. It would be good if the authors could provide a short argument to explain why the P^* in line 507 attains the optimal of line 506. (This is clear to experts but is not immediately obvious for non-expert readers.)

4. Section 3.2 could have been developed in much more detail, for example, by providing some (simple) examples to demonstrate the non-triviality. See also the Question section for more concrete examples.

[BHS23] Bhatt, A., Haghtalab, N., and Shetty, A. (2023). Smoothed analysis of sequential probability assignment. NeurIPS 2023.

[RSS12] Rakhlin, S., Shamir, O., & Sridharan, K. (2012). Relax and randomize: From value to algorithms. NeurIPS 2012.

**Questions:**

Although the contextual Shtarkov sum representation can be used to recover the upper bounds of [BFR20] and [WHGS23], potentially with improved constants, it would be beneficial to see some explicit examples demonstrating the benefits. For example, consider the expert class $\mathcal{F}=\left\\{\frac{\langle w,x\rangle+1}{2} : w \in B_2\right\\}$, where $B_2$ is the unit $L_2$ ball. Assume the contexts are also selected from $B_2$. We know that the bounds in [BFR20] and [WHGS23] can only provide an $\tilde{O}(T^{2/3})$ upper bound. However, it is known from [RS15] that a Follow-the-Regularized-Leader predictor achieves regret upper bounded by $\tilde{O}(\sqrt{T})$.

Can the characterization introduced in the current paper provide an alternative (perhaps simpler) derivation of this $\tilde{O}(\sqrt{T})$ regret?

---

> ### Author Rebuttal · Authors · 2024-08-07
>
> We thank the reviewer for the detailed and knowledgeable review. We appreciate your recognition of our main results, and hope you will help us defend the merit of our work. We first address your comments on the weaknesses of our paper and then answer your questions.
> # Clarification on our technical contribution
> We appreciate that the reviewer appreciates the true novelty of Lemma A.3. It is the true workhorse of our analysis. However, we would also like to clarify some other parts that are novel and not just following the literature. First, the idea of minimax swap is indeed standard in online learning, but for log-loss we need a truncation argument as in the proof of Lemma 6 in [BFR20]. More specifically, we applied a smooth truncation that works for multi-labels. Second, we were able to generalize our characterization of the minimax regret for classes that may not enable minimax swaps, via a novel approximation argument that is also powered by our smooth truncation. Notably, previous results, like Thm 2 in [BFR20] and Lemma 10 in [RS15], implicitly assumed some regularity condition that validated the minimax swap.
>
> We also want to explain a bit about the optimization problem in line 206 and 506. It is basically of the form $\sup_{P} H(P)+ \langle P, x \rangle$ where $P$ ranges over all $d$-dimensional probabilistic vectors and $H$ is the entropy function. The value of this optimization problem is exactly the Fenchel conjugate of $-H$ on $x$, which is the log-sum-exp function $\log(\sum_i \exp(x_i))$, and the maximum is obtained by $P^*$ with $P^*_i=\exp(x_i)/\sum_j \exp(x_j)$.
>
> # On the relation to [BHS23] and the relaxation framework
> Thanks for pointing us to this relevant work of [BHS23]. We will cite it clearly in our final revision. We agree that the spirits of the two papers are similar: the minimax swap in our paper helps reduce the original game to the dual game, and the coupling lemma in [BHS23] helps reduce their problem to transductive learning. A minor difference is that minimax swaps would yield equalities (under certain regularity assumption of the hypothesis class), while the reduction in [BHS23] would produce inequalities.
>
> We also thank the reviewer for raising the question of whether cNML fits into the relaxation framework in [RSS12]. We agree that the worst-case log contextual Shtarkov sum with prefix can be viewed as a (trivially) admissible relaxation, now that we've established that it is exactly the conditional game value. What's nice about this perspective is that we can cite [RSS12] and suggest that (further) admissible relaxations of our contextual sum may yield new algorithms.
> # On section 3.2 and the linear class
> In section 3.2, we connect our new complexity measure and the well-studied sequential Rademacher complexity, through the lens of martingales. The quantitative study of sequential Rademacher complexity greatly benefits from its martingale structure, see for example [RST15]. So we hope to inspire future work on developing new tools suitable for the product-type martingale embedded in contextual Shtarkov sum. Indeed, the linear class is a great example exhibiting the inadequacy of sequential covering numbers as the right complexity measure, so it automatically justifies our new complexity, although currently, we lack the right math tools to give a tight bound for it directly from contextual Shtarkov sum. We believe it is an important future direction to study our contextual Shtarkov sums in a systematic and quantitative way, as has been done for sequential Rademacher complexity.
>
> [RSS12] Rakhlin, Shamir, and Sridharan. (2012). Relax and randomize: From value to algorithms.\
> [RS15]: Rakhlin and Sridharan. (2015). Sequential probability assignment with binary alphabets and large classes of experts.\
> [RST15] Rakhlin, Sridharan, and Tewari. (2015). Sequential complexities and uniform martingale laws of large numbers.\
> [BFR20] Bilodeau, Foster, and Roy. (2020). Tight bounds on minimax regret under logarithmic loss via self-concordance.\
> [BHS23]: Bhatt, Haghtalab, and Shetty. (2023). Smoothed analysis of sequential probability assignment.

---

> > ### Comment · Reviewer_qSjy · 2024-08-07
> >
> > I thank the authors for the detailed response. After reading the rebuttal and checking the feedback from other reviewers, I have decided to confirm my strong accept recommendation. This is a clear significant result that resolves a long-standing problem, and I will argue for acceptance.

---

> > > ### Author Response · Authors · 2024-08-14
> > > **Thank you!**
> > >
> > > Thank you for the encouraging comment. We will revise our paper as suggested.

---

### Official Review · Reviewer_ZwzX · 2024-06-29

**Soundness:** 2
**Presentation:** 3
**Contribution:** 2
**Rating:** 4
**Confidence:** 2

**Summary:**

The authors explore the fundamental problem of sequential probability assignments, specifically focusing on log-loss online learning with an arbitrary, possibly non-parametric hypothesis class. They introduce a new complexity measure and demonstrate that the worst-case contextual Shtarkov sum equals minimal regret. With this, they derive a minimal optimal strategy, called cNML, which extends beyond binary labels.

**Strengths:**

The paper successfully generalizes classical results in sequential probability assignment to contexts that involve multiary labels and sequential hypothesis classes.
Another strength of this paper is its robust characterization of minimax regret and optimal prediction strategies holds for arbitrary hypothesis classes. This is particularly good as it relaxes the assumptions often required in earlier works.

**Weaknesses:**

The paper does not include experimental results to support the theoretical findings.

**Questions:**

Perhaps the authors could conduct some experiments to compare the performance of different algorithms, so as to visually demonstrate the effectiveness of the newly proposed cNML algorithm compared to other algorithms (e.g., traditional NML, Bayesian methods, etc.) on real datasets, especially when dealing with multi-label and complex sequence prediction problems.

**Limitations:**

Yes

---

> ### Author Rebuttal · Authors · 2024-08-07
>
> We thank the reviewer for their constructive review. Below, we address your comments on experimental results and our cNML algorithm.
> # On experimental results and our cNML
> We would like to elaborate on the statement of our results. Theorem 3.2 says that if we compute the worst-case regret of all algorithms, then the minimal worst-case regret that any algorithm can achieve is exactly the worst-case log contextual Shtarkov sum. Then Theorem 4.2 shows that cNML is the exact algorithm that achieves this minimal worst-case regret, meaning that cNML is the minimax optimal algorithm. So if we compare the regrets of cNML and any other algorithm (say any Bayesian method) on their respective worst-case sequences, then cNML will have minimal such regret. Hence, if we really want to witness the gap between cNML and other algorithms, we have to first find out their respective worst-case data sequences (which can be different for different algorithms). In our view, it is not instructive to compare the regrets of algorithms on real datasets since such real data is not guaranteed to be the worst-case data for our algorithms of interest.
>
> We also want to clarify that cNML is a novel and highly non-trivial generalization of the NML algorithm which is suitable for the context-free setting. So basically NML cannot run in the more general and complicated contextual setting that we studied in this paper.

---

### Official Review · Reviewer_upYt · 2024-07-10

**Soundness:** 4
**Presentation:** 3
**Contribution:** 2
**Rating:** 6
**Confidence:** 4

**Summary:**

In this work, the authors consider the problem of sequential probability assignment, also known as online learning with the log-loss.  While earlier results have shown that in the context-free or transductive settings, the Shtarkov complexity is minimax optimal (and attained with normalized maximum likelihood), the contextual analogue was bounded only in terms of sequential covering numbers.  The authors propose a new complexity measure, the contextual Shtarkov sum, that is precisely equal to the minimax regret.  The authors then bound this complexity in terms of the sequential covering numbers in a way that improves marginally on prior work.

**Strengths:**

The paper discusses an important problem (that of sequential probability assignment) and provides a new bound on the minimax regret.  This new bound depends on a new notion of complexity that acts as the analogue of the sequential Rademacher complexity for log loss.  The paper is presented well and explains its technical details clearly.

**Weaknesses:**

The paper's contributions are somewhat incremental.  The Shtarkov complexity is well-known for case of transductive learning and the introduction of contexts uses techniques introduced and used in many earlier works on sequential Rademacher complexity.  This is not helped by the fact that the Shtarkov complexity itself is almost tautalogically equal to the minimax regret.  This weakness would be somewhat mitigated if the new complexity measure led to improved bounds on the minimax regret, but the claimed improvement over BFR20 is only at the level of a constant as, asymptotically as $\alpha \downarrow 0$, the upper bound still scales like $\inf_\alpha \{ 2 T \alpha + \mathcal H_\infty(\mathcal F, \alpha, T) \}$.  Furthermore, on the question of depth of technical contribution, it is not clear to me what is really novel.

**Questions:**

1. Could the authors more clearly explain the novelty in technical contributions in their work?

2.  When invoking the Sion's minimax theorem in the proof of Theorem 3.2, the compactness of the space should be mentioned as it is a necessary condition.  The proof is correct, however.

3. I assume that the $H(P)$ in line 206 is entropy, but I have a hard time seeing where this notation is introduced in the paper.

4. While section 3.2 seems like an interesting interpretation of the Shtarkov sum, it seems like a bit of a non sequitur.  How does it relate to the rest of the paper?

5. Much is made of the fact that the approaches in this paper address the multi-label setting, beyond binary $\mathcal Y$, but it is not clear why this is technically more challenging.  Indeed, do the earlier works of BFR20 not immediately generalize beyond binary $\mathcal Y$?

**Limitations:**

The discussion of limitations is adequate, although mention of the computational challenges of their algorithm even in the case of convex classes (due to the necessity of optimizing over the space of trees, which is exponentially large) would be good to include.

---

> ### Author Rebuttal · Authors · 2024-08-07
>
> We thank the reviewer for the detailed review and, in particular, acknowledging the importance of the problem being studied and praising our presentation. To begin, we address your concern about the novelty in our global rebuttal. We will address your remaining comments on the weaknesses of our paper and then answer your questions.
>
> # On improved bounds on the minimax regret
> To see that our new complexity leads to improved bounds, first we would like to emphasize that the worst case log contextual Shtarkov sum **equals** the minimax regret, which means that on every instance where the bound in [BFR20] is loose, our result will be provably better.
> For example, the linear class {$\frac{\langle w,x\rangle + 1}{2}: w\in B_2 $} and the absolute linear class {$|\langle w,x\rangle|: w\in B_2 $}  (where contexts $x$ are also from $B_2$) have the same sequential covering numbers (up to constants) but admit different rates of minimax regret. Our goal is to come up with a precise characterization (that is not based on covering) to circumvent this negative result. Another benefit from our characterization is to realize the product-type martingale structure embedded in contextual Shtarkov sums, which can motivate the development of new math tools and regret bounds in the future (for more details, see the later discussion on section 3.2).
>
> So overall we don’t aim to follow the bound in terms of sequential covering numbers too closely. Our purpose of giving a quick proof of the bound in [BFR20] (with better constants) is to showcase that starting from our new complexity, it is **easy** to get a covering-based bound as in [BFR20] but with **optimal** constants. See the argument at the bottom of page 6 in [WGHS22] for the optimality of our constants, where their bound is in terms of the global covering number that is no smaller than the local covering number in our bound.
>
> Moreover, our bound is the **first constructive** one (meaning it’s achieved by our cNML algorithm) that is optimal in terms of the local sequential entropy:
> The upper bound on the regret in [BFR20] is non-constructive (no algorithm is provided).
> While [WGHS22; WGHS23] provided a concrete algorithm, their bound depends on the worse global covering number, as opposed to the local covering numbers in our bound.
>
> # On generalizing [BFR20] beyond binary labels​
> First, we want to clarify that the bound in [BFR20] is already loose for some classes. So despite the possible generalization, we don’t expect it to give us a tight bound in the multi-label setting. More specifically,  we can similarly upper bound the regret by an offset process using self-concordance and control this offset process using a nice enough cover. However, there is always an issue with the tightness of such covering-based upper bounds: the one in [BFR20] has been proved to be loose for some cases, and for other Dudley’s entropy-like bounds (e.g. Prop 7 in [RST10] and Thm 1.1 in [DG22]), they don’t admit matching lower bounds as well.
>
>
> # On the presentation
> Thank you for pointing out the missing explanations about the entropy function and the use of Sion’s minimax theorem. In our final revision, we will introduce the notation of the entropy function in the main body and mention the compactness of relevant spaces in the proof of Theorem 3.2.
>
> Regarding our section 3.2, our purpose is to establish a connection between our new complexity and the well-studied sequential Rademacher complexity through the lens of martingales. In this way, we hope to inspire future research that develops new tools suited to the product-type martingale embedded in contextual Shtarkov sums, following a similar path of the quantitative study of sequential Rademacher complexity (see [RST15] for example).
>
> [RST10]: Rakhlin, Sridharan, and Tewari. (2020). Online learning: Random averages, combinatorial parameters, and learnability.\
> [RST15] Rakhlin, Sridharan, and Tewari. (2015). Sequential complexities and uniform martingale laws of large numbers.\
> [DG22]: Daskalakis and Golowich. (2022). Fast Rates for Nonparametric Online Learning: From Realizability to Learning in Games.\
> [BFR20]: Bilodeau, Foster, and Roy. (2020). Tight bounds on minimax regret under logarithmic loss via self-concordance.\
> [WGHS22]: Wu, Heidari, Grama, and Szpankowski. (2022). Precise Regret Bounds for Log-loss via a Truncated Bayesian Algorithm.\
> [WHGS23]: Wu, Heidari, Grama, and Szpankowski. (2023). Regret Bounds for Log-loss via Bayesian Algorithms.

---

> > ### Comment · Reviewer_upYt · 2024-08-10
> > **Thank you for the response.**
> >
> > Thank you for the detailed response.
> >
> > To clarify, my score is not based on the presentation as the issues I raised are minor.  I personally find the observation that the contextual Shtarkov sum can be realized as the expected supremum of a martingale to be quite cool; I simply was making the minor point that the section did not seem to be connected to the rest of the paper in a natural way.
> >
> > On the subject of improved bounds, I would encourage you to include a detailed discussion of the linear vs absolute-linear example in the paper.  What is the difference in the rates of online learning these classes with log loss and just how loose is a covering number or Dudley based bound here?
> >
> > Regarding the global rebuttal, I agree with you that the problem studied is an important one, but I remain somewhat confused as to the specific claimed novel techniques applied and am not yet fully convinced that the assumed regularity properties of the class $\mathcal F$ that are required by earlier cited works could not be avoided with very minor tweaks to the proofs, certainly if one is willing to pay a constant factor on the regret.  I will raise my score on the assumption that the authors will include a detailed discussion of at least one example, such as the linear vs absolute linear case, where their new complexity measure improves on the local covering number based bound beyond constant factors.

---

> > > ### Author Response · Authors · 2024-08-14
> > > **Thank you!**
> > >
> > > Thank you for the clarification. We will incorporate all your suggestions into the final revision.

---

### Official Review · Reviewer_rqqz · 2024-07-15

**Soundness:** 4
**Presentation:** 2
**Contribution:** 3
**Rating:** 6
**Confidence:** 4

**Summary:**

This paper studies minimax regret in the online learning setup. The paper extends prior works to nonbinary labels and proves improved bounds on the regret. The analysis is through studying the Shtarkov sum that characterizes the minimax regret.  The authors develop a data-dependent variant of the contextual Shtarkov sum and introduce a variant of the Normalized Maximum Likelihood. Moreover, the paper takes an information-theoretic approach and studies sequential $\ell_\infty$ entropy terms to give upper bounds on regret.

**Strengths:**

The paper is solid and presents a nice mathematical framework to prove bounds on the regret in online learning. The frameworks allows one to study a broader class of online learning problems including non binary labels. In addition it makes the proof a bit nicer with slightly improved bounds on the regret. Overall I liked the technical contributions of the paper as a framework to study various online learning problems.

**Weaknesses:**

On the negative, side it seems that the novelty of the work is somewhat limited, given the cited prior works. It looks like that the authors extend the Sharkov sum and its analysis to the contextual case.

The presentation of the paper could be better. It starts with elementary definitions and explanations of the problem but jumps to the technical parts without a smooth transition.

Moreover, the seemed that the paper is dry and lacks giving enough motivation and explanation.  For instance, the authors introduce multiary trees without explaining the high-level picture. It is not clear why and how this concept is used in the proofs. As another example, it is not clear at the beginning why the proposed algorithm (cNML) is useful.  Perhaps a discussion on this matter helps the reader kept interested in the paper.

**Questions:**

Has a contextual algorithm similar cNML been introduced before ? If so what are the benefits ?

Any specific example of the multiary tree would be appreciated.

**Limitations:**

Nothing significant.

---

> ### Author Rebuttal · Authors · 2024-08-07
>
> We thank the reviewer for the detailed review and, in particular, your recognition of our technical contributions. To begin, we address your concern about the novelty in our global rebuttal. We will address your remaining comments on the weaknesses of our paper and answer your questions.
> # On cNML
> Indeed, [WGHS23] gave a contextual algorithm based on the global sequential covering, which has been shown to be sub-optimal in some cases. In contrast, our cNML is provably minimax optimal, that is, it attains the optimal regret for any hypothesis class.
> More specifically, if we compare the worst-case regret of all possible algorithms, then our Thm 4.2 shows that cNML will achieve the minimum, which is exactly the worst-case log contextual Shtarkov sum (see our Thm 3.2).
> Moreover, we expect that the cNML will lead to the development of new practical algorithms for the contextual setting, similar to the NML in the context-free case.
>
> Regarding the novelty of cNML, we are not aware of any similar algorithm in the contextual setting from the literature. It’s a novel and non-trivial contextual generalization of NML (established in [Sht87]). Just like NML is minimax optimal in the context-free setting, cNML is minimax optimal in the more general and complicated contextual setting. Above all, we consider designing the cNML algorithm as one of our major contributions.
> # On the presentation
> Thank you for useful suggestions on the improvement of our presentation. We will gladly incorporate them into our final revision.
> We agree that some intuitions behind multiary trees can be helpful. To see how multiary trees show up in the proof, we would like to note that after rewriting the minimax regret via the minimax swap (which is standard in the theoretical online learning papers but needs additional care for the log-loss here), multiary trees arise as a natural notion that can be used to further simplify this new expression of the minimax regret. For example, see section 4.1 of [BFR20] for reference. A direct visualization of a multiary tree with value in the context space $\mathcal X$ is a complete rooted multiary tree each of whose nodes are labeled by an element of $\mathcal X$.
>
>
> [Sht87] Shtarkov. (1987). Universal Sequential Coding of Single Messages.\
> [BFR20] Bilodeau, Foster, and Roy. (2020). Tight bounds on minimax regret under logarithmic loss via self-concordance.

---

> > ### Comment · Reviewer_rqqz · 2024-08-11
> >
> > Thank you for your response. Parts of my concerns have been addressed and I would be happy to increase my score.

---

> > > ### Author Response · Authors · 2024-08-14
> > > **Thank you!**
> > >
> > > Thank you for the positive comment. We are happy to see that some of your concerns are addressed.

---

### Author Rebuttal · Authors · 2024-08-07

Multiple reviewers question the novelty of this work, but also express some uncertainty by not expressing complete confidence. Here we give a full throated defense of our work's significance and novelty. We hope this settles the matter.
# On the significance of the problem
First, we want to highlight the significance of the fundamental problem studied in this paper. The problem of sequential probability assignment (with contexts/side information) was raised no later than 2006 in the online learning "bible" [CL06] (see sec 9.9). Since then, this problem has been studied under the framework of online supervised learning, formalized e.g. by [RST10], with the objective of characterizing the minimax regret w.r.t general hypothesis classes. Prior works (including [RS15; BFR20; WGHS23]) focused on upper bounding the regret by sequential covering number, which turns out NOT to be the right complexity measure: there exist two natural hypothesis classes such that their sequential covering numbers are of the same order, but there is a polynomial gap between their rates of minimax regret (see sec 3.2.2 in [BHS23]). Previous attempts by colleagues to perform a general minmax swap had not succeeded and so the problem of identifying a general complexity measure that characterized minimax regret has stubbornly remained open for nearly 20 years, nevermind a minimax algorithm for general hypothesis classes, which was also missing. This was a key open problem in machine learning theory.
# On our technical contribution
Now we highlight that our technical contribution is non-trivial and novel in the following senses: \
**Novelty of the notion of contextual Shtarkov sums:** Our definition of contextual Shtarkov sum is very much different from those in the non-contextual and transductive settings and serves as a novel generalization. It's important to not lose track of the fact that, after we show that it characterizes the minimax regret, it will of course seem to be the natural and right complexity measure in our contextual and fully adaptive setting. To be more specific, the Shtarkov sum in the non-contextual setting was introduced in 1987 by [Sht87]. It is immediate to derive a Shtarkov sum for the transductive setting and to show that it characterizes the minimax (fixed-design) regret there, since knowing the context sequence to be revealed reduces the problem to the non-contextual setting! This is not the case in the adaptive setting! In contrast, prior to our work, the right generalization of the Shtarkov sum to our setting was entirely absent (for 20 years!). For example, in chapter 9 of [CL06], the traditional Shtarkov sum is extensively exhibited (see Thm 9.1) and the problem of sequential probability assignment with contexts is clearly introduced (see sec 9.9). The long list of researchers who have worked on this problem have all been aware of Shtarkov sums, and several have since reported to us trying this approach, but failing. \
**Novelty of our proof techniques:**
- The techniques we applied to derive that the minimax regret is characterized by contextual Shtarkov sums are very (!) different from those for sequential Rademacher complexity, which have dominated studies of online supervised learning. As reviewer qSjy points out, we are able to characterize the minimax regret by leveraging properties of log-loss, thanks to our novel Lemma A.3. More specifically, we make an important observation that probabilistic trees are equivalent to joint distributions, which further enables us to derive that the expected cumulative loss of learner (after the minimax swap) is actually the entropy of this joint distribution. Therefore, we reduce the minimax regret into an entropic-regularized optimization problem whose value can be computed explicitly. In light of this, we are able to characterize the minimax regret by (the worst-case log of) contextual Shtarkov sums.
- Another non-trivial part of our analysis is the minimax swap. This is due to the fact that log loss is non-Lipschitz and unbounded, unlike absolute and square losses. So we applied a truncation method that works for multi-labels to apply minimax swaps. Moreover, previous works implicitly assumed that minimax swaps always hold and they required a regularity assumption for the hypothesis class ([RS15] for their lower bound and [BFR20] for their upper bound), so their regret bounds are all conditional on this assumption. However, we apply a novel approximation argument to show that contextual Shtarkov sums continue to characterize the minimax regret w.r.t all hypothesis classes that don’t necessarily validate minimax swaps. In this way, we are able to get rid of any regularity assumption.

**Novelty of cNML algorithm:** Our second major contribution is the design of the cNML algorithm, which we show is minimax optimal for sequential probability assignment with contexts. Thus, cNML can be thought of as the contextual generalization of NML (which dates back to [Sht87]) from the context-free setting. Also, cNML is the first minimax algorithm for this problem. Previous ones like the Bayesian algorithm in [WGHS23] only admit suboptimal regret guarantees. Moreover, we believe cNML is the starting point of more practical algorithms in future works, just like NML.

[Sht87] Shtarkov. (1987) Universal Sequential Coding of Single Messages\
[CL06] Cesa-Bianchi & Lugosi. (2006) Prediction, learning, and games\
[RST10] Rakhlin, Sridharan, & Tewari. (2020) Online learning: Random averages, combinatorial parameters, and learnability\
[RS15] Rakhlin & Sridharan. (2015) Sequential probability assignment with binary alphabets and large classes of experts\
[BFR20] Bilodeau, Foster, & Roy. (2020) Tight bounds on minimax regret under logarithmic loss via self-concordance\
[BHS23] Bhatt, Haghtalab, & Shetty. (2023) Smoothed analysis of sequential probability assignment\
[WHGS23] Wu, Heidari, Grama, & Szpankowski. (2023) Regret Bounds for Log-loss via Bayesian Algorithms

---

### Decision · Program_Chairs · 2024-09-25

**Decision:**

Accept (poster)

**Comment:**

The paper studies the minimax regret in the online adversarial setting (with contexts) when one uses log loss and arbitrary class of functions/predictors. They introduce a complexity measure they refer to as contextual Shtarkov sum that provides a minimax regret bound for the problem in the multi alphabet outcome case. The paper improves upon previous results in [RS15],  [BFR20] and [WHGS23] and provides the right rate for the problem. The work is definitely worth publishing at Neurips. Majority of the reviewers agree with this view.